



# Data-driven estimates for the geostatistical characterization of subsurface hydraulic properties

Falk Heße[1,2], Sebastian Müller[2,1], and Sabine Attinger[2,1]

[1]Institute of Earth and Environmental Sciences, University of Potsdam, Karl-Liebknecht-Str. 24–25, 14476 Potsdam, Germany,
[2]Department of Computational Hydrosystems, Helmholtz Centre for Environmental Research – UFZ, Permoserstr. 15, 04318 Leipzig, Germany

**Correspondence:** Falk Heße (falk.hesse@ufz.de)

**Abstract.** The geostatistical characterization of the subsurface is confronted with the double challenge of large uncertainties and high exploration costs. Making use of all available data sources is consequently very important. Bayesian inference is able to mitigate uncertainties in such a data scarce context by drawing on available background information in form of a prior distribution. To make such a prior distribution transparent and objective, it should be calibrated against a data set containing

estimates of the target variable from available sites. In this study, we provide a collection of covariance/variogram functions of the subsurface hydraulic parameters from a large number of sites. We analyze this data set by fitting a number of widely used variogram model functions and show how they can be used to derive prior distributions of the parameters of said functions. In addition, we discuss a number of conclusions that can be drawn for our analysis and possible uses for the data set.

## 10  1  Introduction

Due to high exploration costs, the field of subsurface hydrology is characterized by scarcity of data, leading to high uncertainty (Heße et al., 2019). Collecting data and making them available to practitioners should therefore be a high priority. In the field of subsurface hydrology, the largest data bases are the World Wide Hydrogeological Parameters DAtabase (WWHYPDA) (Comunian and Renard, 2009) for aquifer data as well as the SoilKsatDB for soil data (Gupta et al., 2021). These data bases

were launched in 2006 and 2021 with the aim of creating a collaborative catalog of values and statistical distributions needed for subsurface hydrological modeling. The data are stored together with metadata like estimated measurement errors, number of metadata on the site, the measurement technique, length scale, rock or soil type, etc.

As such, they can serve as a repository for background information, that practitioners can draw on to improve their understanding and modeling of the subsurface. Bayesian inference is known for being able to incorporate such background

information by virtue of the prior distribution and therefore provide information for free. While the role of priors and their choice in statistical inference used to be strongly debated, it is now widely acknowledged that priors that are based on trans-





parent, impartial and observable base rates provide an objective source of information (Billot et al., 2005; Gilboa et al., 2010; Gelman and Hennig, 2015). In a data scarce context such a as subsurface hydrology, being able to draw on such a source of free information is an invaluable asset which has not been fully used so far (Heße et al., 2019). Recently, Cucchi et al. (2019)

developed and introduced a Bayesian hierarchical model that addresses parts of this challenge. Using this model, it is possible to derive prior distributions for one-points statistics like mean and variance (Heße et al., 2021). One of the main challenges, however, is the ongoing lack of data on higher-order statistics which would allow to derive prior distributions for models that describe spatial correlations like the covariance or (semi-)variogram function. Examples of such statistics are the horizontal and vertical correlation/integral scales, anisotropy ratios and variogram/covariance models. As a result, no open-access tools which

systematically provide background information on such variables are available. This means that even a simple structural model for spatial heterogeneity, like a Gaussian process (Gelfand and Schliep, 2016), is currently lacking objective and informative prior distributions for its main parameters. To address this, it is necessary to collect and analyze a data set both large enough and suitable for statistical analysis, that will help to specify prior distributions of such multivariate parameters.

For the collection of these data, different sources are available: primary data in the form of geo-referenced point measure-

ments, secondary data in the form of empirical variogram functions, and tertiary data in the form of statistical estimates of subsurface properties. As regards primary data, the SoilKsatDB database provides some geo-referenced measurements, while the WWHYPDA unfortunately does not. In addition, the research literature provides a substantial yet disorganized repository on such data (Bjerg et al., 1992; Rehfeldt et al., 1992; Hess et al., 1992; Welhan and Reed, 1997; Vereecken et al., 2000), primarily for conductivity and transmissivity fields. As regards secondary data, a large number of empirical variogram clouds can

be found in the literature. In fact, they provided the majority of estimates on higher-order statistics for our study (see below). In addition, some sources provide curated collections of tertiary data in the form of subsurface statistics, which can be used directly (Jim Yeh, 1992; Gelhar, 1993; Kupfersberger and Deutsch, 1999; Rubin, 2003).

Apart from its above mentioned value for Bayesian inference, a large data set of spatial correlations can be important for a wide range of applications and investigations. First, geo-statistical subsurface parameters like the characteristic length scale

(Neuman, 1990; Rovey II and Cherkauer, 1995; Sanchez-Vila et al., 1996; Schulze-Makuch et al., 1999; Bromley et al., 2004) or the dispersion coefficient (Pickens and Grisak, 1981; Arya et al., 1988; Cirpka and Kitanidis, 2000; Dentz et al., 2011; Ross et al., 2019) are widely known to show scale effects. This effect is such that their estimated value increases with the observation scale. This observation is used to argue that the subsurface should be characterized as a fractal medium (Neuman et al., 2008). Yet so far, this scale dependency has mostly been investigated theoretically or using small data sets (Zech et al., 2015). With a

data set like the one provided here, the community of subsurface geostatistics has an empirical basis to investigate this question in more detail.

Furthermore, the data set can be used to compare different established variogram models by, for example, investigating how they differ in parameter estimation with respect to length scale or nugget effect. Furthermore, some variogram models have additional shape parameters. A large data set can be used to determine how such added complexity can help to better describe

empirical variogram functions and whether the added complexity is justified by greater accuracy in modeling.



Even outside of Bayesian parameter estimation, a data-driven approach like ours can be of use for classic parameter estimation. Virtually all geo-statistical software tools provide the ability to use user-specified initial values. Having good initial values can be key in any optimization routine and our results can provide such estimates.

Finally, the data set provided here can be used as an empirical basis for a wide range of investigations into the properties and and characteristics of subsurface quantities like hydraulic conductivity. Such additional studies can, e.g., investigate under which circumstances any given variogram model function is the best choice, whether there is any connection between a given property of the experimental variogram and some other property of the underlying medium, it can be used to test the applicability of a new variogram model, to test new hypotheses regarding subsurface behavior, or to investigate whether cross correlations between different parameters exist.

To explain how we addressed the above aims, the remainder of this manuscript is organized as follows: In the next section, we will begin by presenting the methods used in this study to derive our results. This comprises the sources for our data set and how we compiled them together, the covariance/variogram model we used to analyse these data, the software tools and work flow we used as well as the online repositories where all data and software solutions can be accessed. Next comes the results section, where we present and analyze the statistical properties of the different variogram parameters and how they can be used to improve sub-surface characterization. In addition, we will critically assess the limits of our study and discuss dangers of misuse. In the final conclusions section, we close with a summary of our main findings and how practitioners can benefit from them.

## 2 Methods

Let us start by looking at the tools and methods that we used in this study to derive our conclusions. This comprises the data sets on subsurface variogram data, the variogram models that we used to analyze these data and the numerical tools for the analyses.

### 2.1 Data set

#### 2.1.1 Data sources

To obtain a representative data set of subsurface variogram functions, we conducted a literature search on the ISI Web of Knowledge (https://clarivate.com/webofsciencegroup/solutions/web-of-science/, last accessed on 31 August 2022). We searched for these data by using the phrases "hydraulic conductivity", "saturated hydraulic transmissivity", "hydraulic transmissivity", "hydraulic permeability", "correlation length", "spatial variability", "variogram", "semivariogram", "Kriging" and "covariance". We looked at all references that resulted from that search. If it contained subsurface measurements or a geostatistical analysis of them, we added them to the data set. If references were made to available data, we tried to contact the corresponding author(s) of the study. The acquired data can be classified into three main categories, namely: (i) existing data on hydraulic conductivity, transmissivity or permeability (in the form of tables) published in peer-reviewed publications, (ii) processed data





on variogram functions in the form of empirical variogram functions, and (iii) collections of estimated variogram parameters. It is clear that the first form of data is the most useful since it contains little additional processing, whereas the last form represents the least amount of information. Overall, the second form of data were the most common, however.

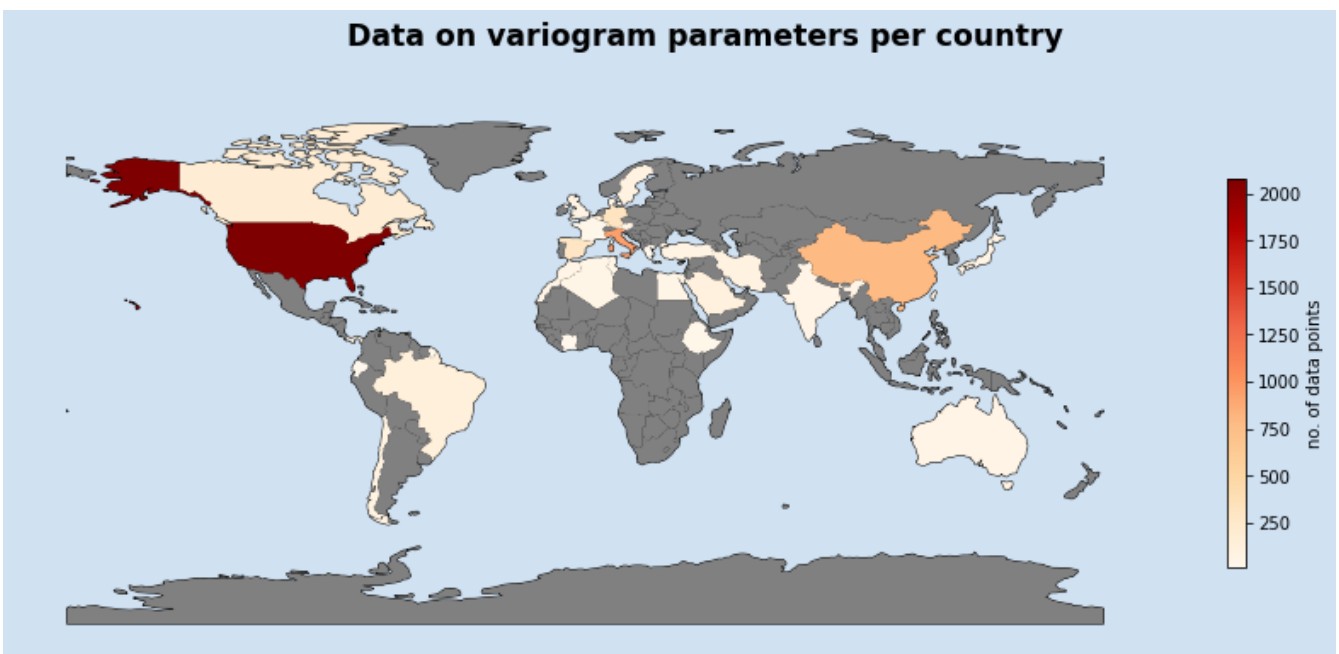

**Figure 1.** Overview of the countries from where data sets were available.

Figure 1 shows a world map of countries where data sets could be collected. The color map here indicates how many data points are available in each country, while countries without data are shown in gray. As can be seen, the focus is on North America and Western Europe, as is common with scientific data. However, other world regions are also covered to a reasonable extent. The data set therefore contains a wide range of climate regions and geographic media.

### 2.1.2   Preparation of the data

Depending on the type of data, we used a number of different workflows to process them. Raw data of hydraulic conductivity, transmissivity and permeability were processed by deriving the empirical variogram cloud, which was subsequently joined with the ones derived from the literature. The empirical variogram clouds found in the literature were available as scatter plots. They were digitized using the freely available WebPlotDigitizer version 4.6 (Rohatgi, 2022). All empirical variogram clouds were then fitted to one of a number of variogram model functions. These model functions and the workflow will be explained

below. The last type of data were processed statistics provided in scientific papers of text books. To avoid any overlap, we made sure that these statistics were not derived from sites which were already present in the other data. For all data derived from the



literature we provide the online sources from where they were taken, by virtue of their digital object identifier in the data file (see below).

### 2.1.3 Representation of the data

All the data used for this study are made available online in a number of `.csv` files. In this section, we are going to describe the keywords of these data files.

**Table 1.** Name, description and units of the key variables used in the results data file

| Header | description | units |
| --- | --- | --- |
| site_id | unique id for the site | – |
| var | estimated sill | – |
| len_scale | estimated length scale | m |
| nugget | estimated nugget | – |
| nu | estimated shape parameter | – |
| r2 | goodness of fit measure | – |
| maximum_scale | maximum length scale of the data set | m |
| minimum_scale | minimum length scale of the data set | m |
| var_type | type of data | – |
| direction | physical direction of the variogram | – |
| geological_unit | specifies possible subunits per site | – |
| data_source | DOI of the data source | – |
| ISO 3166 | country code of the site | – |

These keywords are depicted in Table 1. The first one is `site_id` which provides a unique identifier for the site from which the data were draw. This name is always based on the name used by the authors which collected the data. The next keywords all refer to estimated variogram parameters. These are `var`, `len_scale`, `nugget` and `nu` for the variance, length

scale, nugget and shape parameter, respectively. The shape parameter is not found in all investigated variogram models. In those cases, the entry is empty. The keyword `r2` is the goodness of fit measure, i.e., a measure describing how well a given optimum fit of a variogram model function is actually fitting an empirical variogram cloud. The keywords `maximum_scale` and `minimum_scale` describes the maximum and minimum length scale assumed to be present in the data set. In this study, these length scales are interpreted to represent the largest and smallest distances in the data set. The keyword `var_type`

describes the type of variable. In this study, the data can refer to hydraulic conductivity, saturated hydraulic conductivity (for soil sites), hydraulic transmissivity, hydraulic permeability as well as indicator variograms of hydraulic conductivity. The keyword `direction` describes the direction in which the variogram was taken. Direction `x` is default direction. This means that it was used in cases where a unidirectional variogram was analyzed, and it was used as the main direction when two





horizontal directions were present in the collected data. If two horizontal directions are present, the second direction is always encoded as `y`. Both `x` and `y` have therefore no further physical meaning beyond that. The direction `z` is always used for the vertical direction. The keyword `geological_unit` is used in those situations where several variograms are presented in a source for a given site. This situation can represent a number of different situations. In some cases, the authors of the study separated the data by different geologic strata; in some cases, the separation represented geologic subunits that were subdivided by the authors according to their expertise; in some cases, the data represented several actually distinct sites that were combined into a single measurement campaign; and in some cases, it was not clear what criterion was used to make the separation. This keyword may, therefore, represent a number of different situations. The keyword `data_source` contains the digital object identifier (DOI) to the online resources from which the data were draw. Finally, the keyword `ISO 3166` contains the country code for the country where the data were collected.

## 2.2 Variogram models

In this study, we used the GSTools Python package (Müller et al., 2022) for the analysis of the empirical data and the covariance models implemented in this package. In total, the data were analyzed with five different model functions, namely the Gaussian function, the Exponential function, the Spherical function, the Matérn function as well as the Stable function. However, in the vast majority of studies the spatial heterogeneity in the subsurface is expressed using the (semi-)variogram function $\gamma(h)$, which is related to the correlation function $\rho(h)$ through the following relationship

$$\gamma(h) = n + \sigma^2 \left(1 - \rho(h)\right).$$

Here, $\sigma^2$ is the variance, $h$ is the lag, i.e., the distance between two observation points, and $n$ being the nugget value. Closely related is the also well know covariance function $C(h) = \sigma^2 \rho(h)$. Such a transformation is possible for all considered variogram/covariance models since they all represent weakly stationary (spatial) processes, meaning that their variance is finite. The first variogram model considered here is the Gaussian model function (Webster and Oliver, 2007). It is defined as

$$\gamma(h) = n + \sigma^2 \left(1 - \exp\left(-\frac{h^2}{\ell^2}\right)\right),$$

with $\ell$ being the characteristic length scale. Next is the Exponential model (Webster and Oliver, 2007), which is defined as

$$\gamma(h) = n + \sigma^2 \left(1 - \exp\left(-\frac{h}{\ell}\right)\right),$$

with the parameters having the same definition as above. The next variogram model used is the Spherical model (Webster and Oliver, 2007), which is defined as

$$\gamma(h) = \begin{cases} n + \sigma^2 \left(\frac{3}{2}\frac{h}{\ell} - \frac{1}{2}\frac{h^3}{\ell^3}\right) & h \leq \ell, \\ n + \sigma^2 & h > \ell. \end{cases}$$





These first three model functions are widely used. For example, they represent the vast majority of model functions used in the literature that we used to collect our data set. They also all contain the same number of parameters having the same interpretation. In addition, we also examined two other model functions, both of which contain an additional parameter. The first one is the Matérn function (Rasmussen and Williams, 2005), which is defined as

$$\gamma(h) = n + \sigma^2 \left( 1 - \frac{2^{1-\nu}}{\Gamma(\nu)} \cdot \left(\sqrt{\nu} \cdot h\right)^\nu \cdot \mathrm{K}_\nu \left(\sqrt{\nu} \cdot h\right) \right).$$

Here, $\Gamma$ is the Gamma function and $\mathrm{K}_\nu$ is the modified Bessel function of the second kind (Abramowitz et al., 1972). The $\nu$ parameter sets the Matérn function apart from the above model functions by introducing an additional degree of freedom and therefore more flexibility in modelling the variogram behavior. The final variogram model being used is the Stable model (Wackernagel, 2003), which is defined as

$$\gamma(h) = n + \sigma^2 \left( 1 - \exp\left( -\frac{h^\alpha}{\ell^\alpha} \right) \right).$$

As can be seen, the Stable model, named after the Stable distribution (Wackernagel, 2003), is a generalization of the aforementioned Gaussian and Exponential model by virtue of turning their fixed exponent into the parameter $\alpha$. Even though it is not immediately obvious from its formula, the Matérn function, too, is a generalization of the Gaussian and Exponential model and the additional parameters $\nu$ and $\alpha$, therefore, share some similarities. This will be explored in more detail in the Section 3 below.

These different variogram models were used by us for fitting them to every available empirical data set we collected from the literature. An example is depicted in Figure 2, where four of the five model functions can be seen fitted against an empirical variogram data set of saturated hydraulic conductivity. The specific example data set was collected at an experimental plot site of the Tokyo University of Agriculture and Technology (TUAT), Japan during a measurement campaign in summer 2003 (Wijaya et al., 2010). The resulting best-fit parameters resulting from a fitting procedure like this formed the basis of the following analysis. It should be noted that not all empirical variogram data could be fitted with all five variogram model functions. In those cases where comparisons between different model functions were made, we, therefore, restricted our analysis to those sites for which we could achieve satisfying results for all variogram model functions.

A crucial property of variogram models is the *roughness information* that is closely related to the mathematical concept of differentiability. The roughness $\alpha$ of a correlation function $\rho(r)$ can be defined as (Wu and Lim, 2016)

$$\rho(r) \approx 1 - k \cdot r^\alpha \quad \text{as} \quad r \to 0,$$

with $0 < \alpha \le 2$ and $k > 0$. Low values of $\alpha$ indicate a Gaussian process (not to be confused with the Gaussian variogram model) whose fields are very rough, whereas higher values indicate a process whose fields are very smooth.

The Gaussian model has a roughness information of $\alpha = 2$, the Exponential and Spherical models have $\alpha = 1$, the shape parameter of the Matérn model is directly connected to its roughness information with $\alpha = \min(\nu, 2)$, and in case of the Stable model the shape parameter coincides with its roughness information.





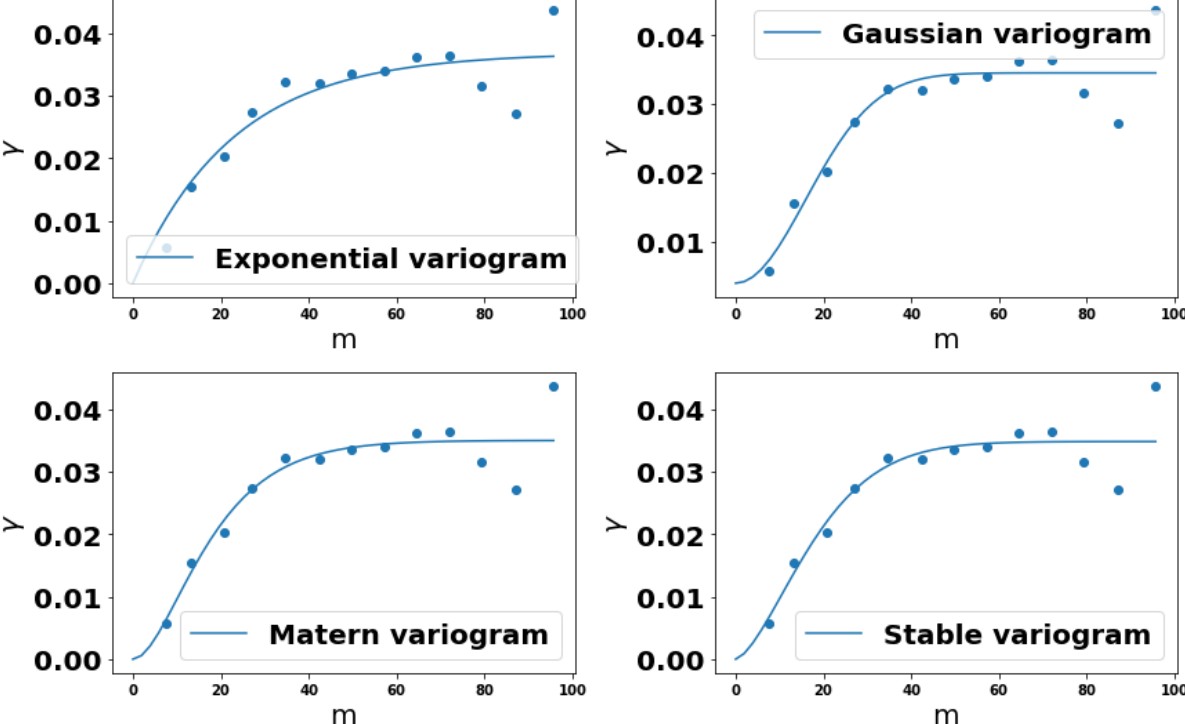

**Figure 2.** Scatter plot of empirical variogram function for saturated hydraulic conductivity presented by Wijaya et al. (2010), jointly with optimal fits using the Gaussian, the Exponential, the Matérn and the Stable model function.

## 2.3  Numerical tools

As mentioned above, the data came from a variety of sources with the majority being from scatter plots of empirical variogram functions presented in scientific articles and reports. In a first step, we digitized them using the freely available WebPlotDigitizer version 4.6. These data were joined into two different `.csv` files, one for aquifer sites and for soil sites. These two data files of the extracted data are available online in the associated GitHub repository, which can be found at GeoStat-Examples/GeoStat-DB/ and is part of the collections of geostatistical examples of the GeoStat-Framework Python packages. The repository contains the whole workflow that generated all the results presented in the paper and ensures transparency and reproducability of the work flow. This is particularly important since we consider the availability of the data set and the prior distribution of certain subsurface parameter to be a key asset of our study. Making all data, results and the work flow that connects them available is therefore mandatory.

A schematic depicting the folder structure can be seen in Figure 3. The `data_raw/` folders contains raw data files, i.e., data on point-referenced measurements of hydraulic conductivity, transmissivity and permeability. Since some of the authors we contacted raised concerns about data ownership, we could not make all raw data available. In those cases, only the empirical variogram data are made available. The `data_prep/` folders contains data on empirical variogram clouds.





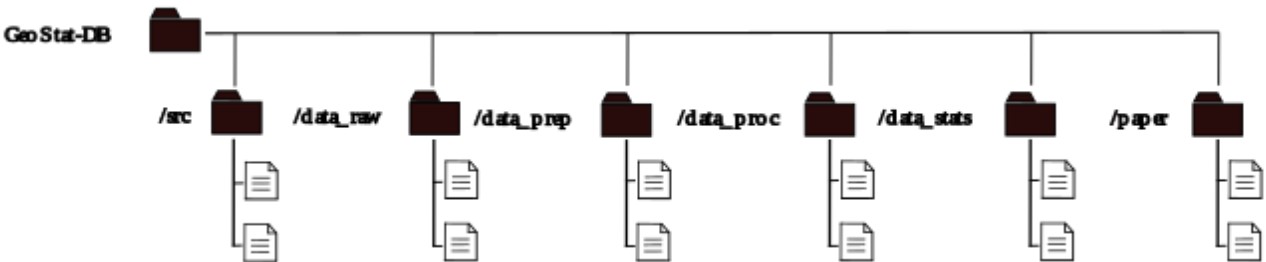

**Figure 3.** Schematic of the folder structure in the GitHub repository associated with this manuscript.

These are either derived from the aforementioned raw data or by digitizing scatter plots from journal articles and reports. The `data_proc/` folders contains processed data, where all empirical variogram data are stored in a single file for convenience. There are two files, one for data from aquifer sites and one for data from soil sites. They form the basis for all the analysis presented below. The `data_stats/` folder contains the results of the geostatistical analysis, i.e., a number `.csv` files that contain the best-fit, geo-statistical estimates. Here, the `.csv` follow the a naming convention such that the

file `aquifer_statistics_gaussian.csv` contains statistics from aquifer sites derived using the Gaussian variogram model, the file `soil_statistics_matern.csv` contains statistics from soil sites derived using the Matérn variogram model, and so on. In addition, this folder contains statistics derived from the literature. The `scr/` folder contains the Python scripts to perform these geostatistical analyses. These files are different depending on whether they are used to analyze aquifer and soil sites and what variogram model is used for the analysis. The file `empirical_aquifer_analysis_gaussian.py`

uses data from aquifer sites and the Gaussian variogram model, the file `empirical_soil_analysis_matern.py` uses data from soil sites and the Matérn variogram model, and so on. The folder also contains the subfolder `geostat_db_tools/`, where Python subroutines that are shared by all the other scripts are placed. Finally, the `paper/` folder contains all data used in the production of this manuscript. This comprises the `paper.tex` file for the main text, the `paper.bib` file for the used references, the figures, and all the scripts used to generate the figures from the results in the `data_stats/` folder. As such,

this repository contains all data and the complete work flow to generate, check on, and improve upon the results present herein.

## 3 Results and discussion

In the following, we will present and discuss the results of analyzing the above data set using the tools introduced in the previous section. To that end, we will focus on the statistical properties of the estimated variogram parameters. These are in particular the length scale, vertical and horizontal anisotropy, the nugget as well as potential shape parameters of the variogram

model function. In addition, we will investigate and compare how different model functions are able to describe empirical variogram data. Since some variogram models have more parameters, i.e., degrees of freedom, we will investigate whether these additional degrees of freedom result in better fitting performance.





### 3.1 Comparison between different variogram model functions

Let us start with a comparison between the different variogram model function using a goodness of fit criterion. The investigated
model functions were the Gaussian model, the Exponential model, the Spherical model, the Matérn model, as well as the
Stable model function. As the goodness of fit criterion, we chose the (pseudo-)$R^2$ measure, also known as the coefficient of
determination, as implemented in the GSTools Python package. In this context, the (pseudo-)$R^2$ score indicates how much
better a fitted model matches the data compared to a pure nugget model set to the mean value of the empirical variogram cloud.

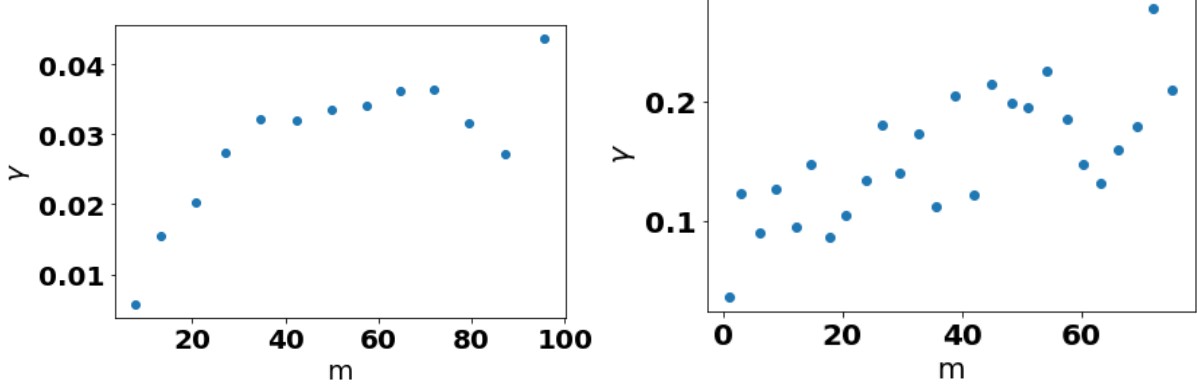

**Figure 4.** Scatter plot of empirical variogram function for saturated hydraulic conductivity presented by Wijaya et al. (2010) and Huysmans
and Dassargues (2006).

Since not all model functions could provide a fit for all sites in the collected data set, we only used those sites for the
comparison where the fitting procedure converged for all considered model functions. Generally, results between the different
model functions varied the most when no clear plateau was reached within the covered spatial range (see right panel in Figure
4). This phenomenon will be also discussed in the following sections where we will look in more detail into the behavior of
different parameters of the model functions.

In general, our results showed comparable goodness of fit measures for all investigated variogram model functions (see
Figure 5). Given that both the Matérn and the Stable model function have one additional degree of freedom and therefore
more flexibility to match any given point cloud, the use of these model functions is not entirely justified by a moderate gain in
accuracy. However, as will be shown and discussed below, the overall similar accuracy of the Gaussian, the Exponential, and
the Spherical model may be a result of the nugget value compensating for some of their lack in flexibility which is restricted
to the area of the curve near the origin. Given that the nugget value isn't a pure convenience parameter but has a plausible
physical interpretation, this behavior of the fitting procedure may be a liability depending on the modeling task.

It is known from the literature that the impact of the specific variogram model function on flow and transport simulations
is mixed (Riva and Willmann, 2009; Jafarpour and Tarrahi, 2011; Heße et al., 2015). Given the overall similar accuracy, these

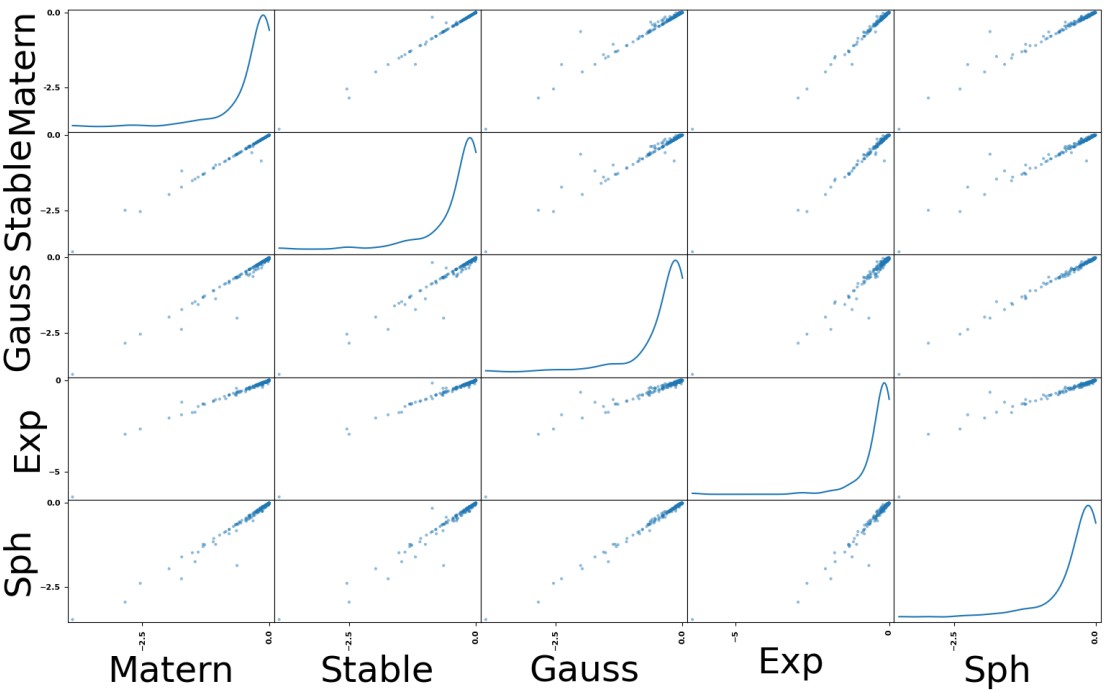

**Figure 5.** Scatter matrix plot for the estimated R2 values using the Gaussian, the Exponential, the Spherical, the Matérn and the Stable model function.

results can interpreted such that the choice of what model function to use for any given task in subsurface hydrology can be driven by considerations of practicality and the specific aims of the task at hand.

One notable difference between the model functions was the number of sites for which our fitting procedure converged and consequently provided usable results (data not shown). The trend was such that the Stable model showed the best performance whereas the Matérn model showed the worst, with the other models being in-between (data not shown). However, this study does not aim to present a thorough analysis of the numerical properties of the different model functions since these often depend on the specific implementation of the model functions themselves, the used functions provided by other packages as

well as the specific set up of the fitting procedure. Using another software or tweaking the fitting procedure can therefore lead to difference in the observed behavior. We would consequently regard these observed differences as tentative and context specific. Regardless, in the following we will use results derived with the Stable model as the default model, when investigating the behavior of specific parameters, if not specified otherwise.





## 3.2 Scale-dependency

As we already discussed in the introduction, the scale dependency of hydraulic properties like the correlation length is a well-know phenomenon from the literature (Neuman and Di Federico, 2003; Neuman, 2008; Colecchio et al., 2020). We therefore investigated this property empirically, by using our data set and estimating correlation lengths for all sites in the data set. The resulting set formed the basis for the following analysis. As mentioned above, we will only present results derived from using the Stable model. This was largely unproblematic since the overall trend for most estimated parameters was the

same regardless of the used model function. Only in cases where some notable changes were observed or where we compare differences between them, do we discuss them separately.

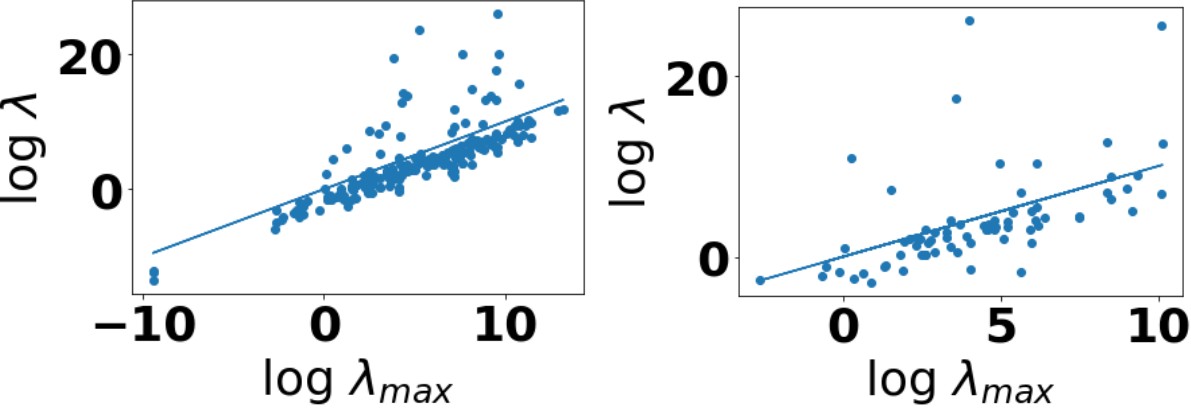

**Figure 6.** Log length scale vs. log maximum length for variogram models fitted to data from aquifers (left) soil (right). The used variogram model function was the Stable model.

Our results confirmed a monotonous increase in correlation lengths with the maximum length scale both for the case of soil and aquifer variogram functions (see Figure 6). Using a log-log plot, we can clearly see an excellent linear relationship between both in the data set. As stated in the methods section, the maximum length scale was defined here as the largest distance in the

data set. In this study, this was identified with the largest distance between two observation points in the data set, typically two piezometer stations or observation wells. We also performed the same analysis with respect to the minimum length scale, which was identified with the smallest distance between two observation points in the data set. As expected, these results showed the same trend (data not shown).

It is not the purpose of this paper to enter into the longstanding debate about the nature of scaling effects and whether

hydraulic variables represent intrinsic physical properties or whether they are introduced only by the measurement process. It can be said, however, that these data, and in particular the striking smoothness of the scaling behavior, provide strong evidence for the notion that the length scale of variogram functions is not primarily an intrinsic physical property of the medium, but is rather influenced by truncation effects induced by the measurement process.





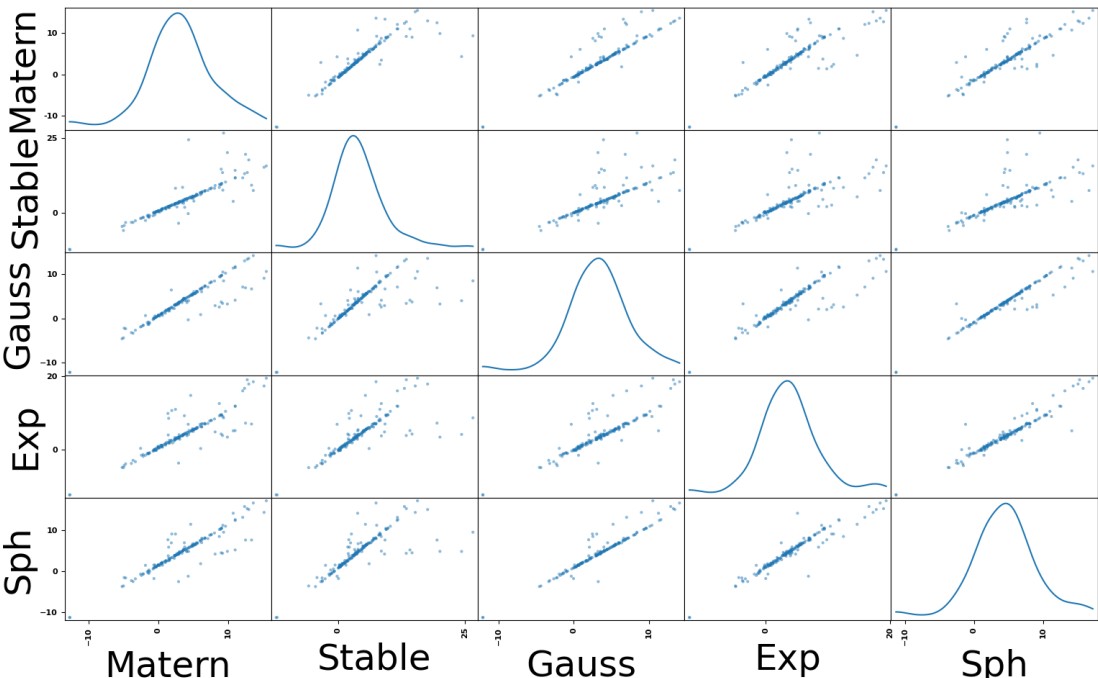

**Figure 7.** Scatter matrix plot for the estimated log length scales using the Gaussian, the Exponential, the Spherical, the Matérn and the Stable model function.

To further investigate the behavior of the estimated length scale, we also looked at the different estimates derived using
different variogram models; namely the Gaussian, the Exponential, the Spherical, the Matérn and the Stable model function.
Results showed an overall strong linear correlation between between the estimates for all investigated variogram models (see
Figure 7). While the slope of the regression plot varied, the overall trend was the same regardless of the used model. This
demonstrates that all models measure the same underlying property of the empirical variogram cloud. Besides this strong
linear correlation, a noticeably number of sites were outliers from this trend, such that they resulted in strongly diverging
estimates depending on the model. We took a closer look at a number of these sites and in all investigated cases we found an
empirical variogram function which had not yet reached a clear plateau. This resulted in a low sensitivity during the fitting
procedure, since only a portion of the expected full variogram behavior was present in the empirical variogram cloud. The
different variogram models therefore reacted differently when exposed to these data and provided sometimes strongly diverging
estimates for those parameters most sensitivity to the long term behavior of the variogram function, namely the length scale
and the variance. It should be noted that in the literature, we found a tendency to perform the fitting such that the plateau of the
model function was reached within the given empirical variogram cloud, probably by enforcing additional constraints during





the estimation procedure. Within this study we did not enforce such conditions resulting in the observed divergence between the different models.

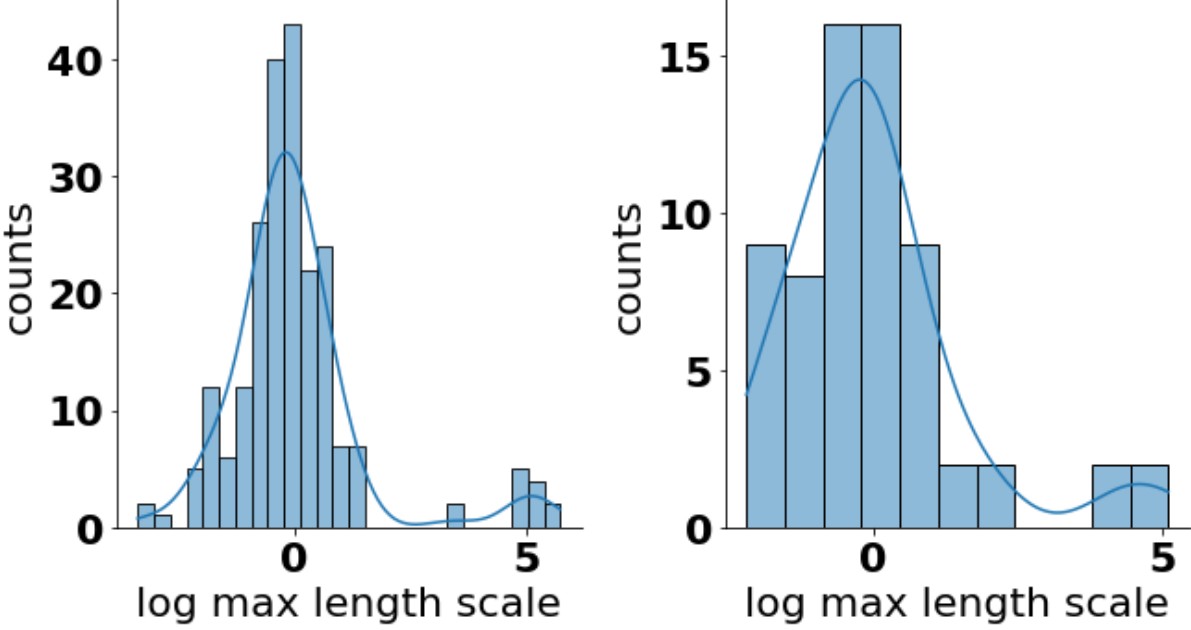

**Figure 8.** Histogram and kernel-density estimate of the residuals around the regression line of the data presented in Figure 6 for aquifers (left) soil (right).

To analyse the behavior of the scale dependency in more detail, we performed a kernel-density estimation for the residuals
around the linear regression line presented in Figure 6. The results showed a similar behavior for both aquifer and soil sites (see Figure 8). In both situations, we saw that most estimated length scales were concentrated at around 1/10th of the maximum length scale with a noticeable uncertainty around that. This value coincides well with an empirical rule of thumb provided by Neuman et al. (2007). Apart from this center of mass, both aquifer and soil sites show estimated length scales that are larger than the maximum length scale present in the data set, a finding that is not explainable by a truncating process. These length
scale estimates which exceed the maximum length scale are not only substantially less common, their estimated value is also much less certain. This is due to the already mentioned fact that only a portion of the overall empirical behavior could be used for the fitting process making the fitting procedure less stable.

All the above results present the length scale determined by fitting a Stable variogram function to the empirical variogram cloud. However as discussed above, the correlation between the estimated length scale was high for all investigated variogram
models. Using another model function consequently resulted in a very similar behavior (data not shown).

From a Bayesian perspective, the distributions of the residuals presented in Figure 8 represent the uncertainty of a length scale estimate given a maximum length scale as a predictor. They are therefore a natural choice for the prior distribution of a

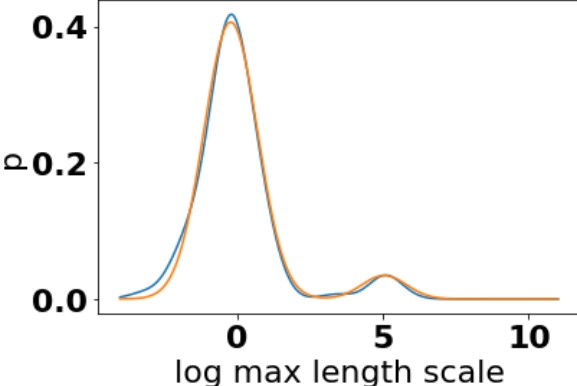

**Figure 9.** Kernel-density estimate (blue) and fitted parametric model (orange) of the residuals around the regression line of the data presented in Figure 6 for aquifers.

Bayesian approach to variogram parameter estimation. Let us demonstrate this approach using the following steps. First, one has to determine the regression for all sites in the data set for the variogram model one wants to use. Let us use aquifer sites only and the Gaussian model function since this is a widely used model. The regression model for the log correlation length given the maximum log length scale then results in

$$log\lambda = 0.99722log\lambda_{max} - 1.355.$$

Here $\lambda_{max}$ would be said maximum length scale, i.e., the predictor of $\lambda$. This represents the knowledge one has regarding the expected correlation length. In the next step, one has to estimate the distribution of the residuals. This represents the uncertainty one has regarding the expected correlation length. In our case, we used a parametric model; namely a mixture model consisting of two independent Gaussian distributions

$$p(log\lambda_{max}) = \frac{\theta}{\sigma_1^2\sqrt{2\pi}}e^{-\left(\frac{log\lambda_{max}-\mu_1}{\sqrt{2}\sigma_1}\right)^2} + \frac{1-\theta}{\sigma_2^2\sqrt{2\pi}}e^{-\left(\frac{log\lambda_{max}-\mu_2}{\sqrt{2}\sigma_2}\right)^2}.$$

Fitting this parametric model yielded the following estimates: $\mu_1 = -0.211$, $\sigma_1^2 = 0.918$, $\mu_2 = 5.048$, $\sigma_2^2 = 0.760$, and $\theta = 0.934$. The goodness of the fit using these parameters can be seen in Figure 9 indicating an excellent representation of the estimated density. This model, i.e., the regression and the prior distribution, can be used by a practitioner for a Bayesian geostatistical modelling of an unknown site.

The above example is, of course, highly contingent on a number of factors. As already mentioned, using a different variogram model may lead to somewhat different estimates and maybe another parametric model may represent the inferred distribution more satisfactorily. Plus, the used data set may change over time or another clustering of the data may lead to different base-rate data sets. Regardless, the above example is a proof-of-concept on how to make use of the assets provided in this study. Of





course all the scripts used to derive above results are available jointly with this manuscript. Practitioners are, therefore, free to
redo the analysis, to check its results as well as adapt them to their needs and applications.

### 3.3  Anisotropy

After having described the scale dependency of the correlation length, let us look at the ansiotropy of these estimates. As it is
well known, subsurface anisotropy is well pronounced between the vertical and horizontal direction but is often assumed to be
negligible between the two horizontal direction.

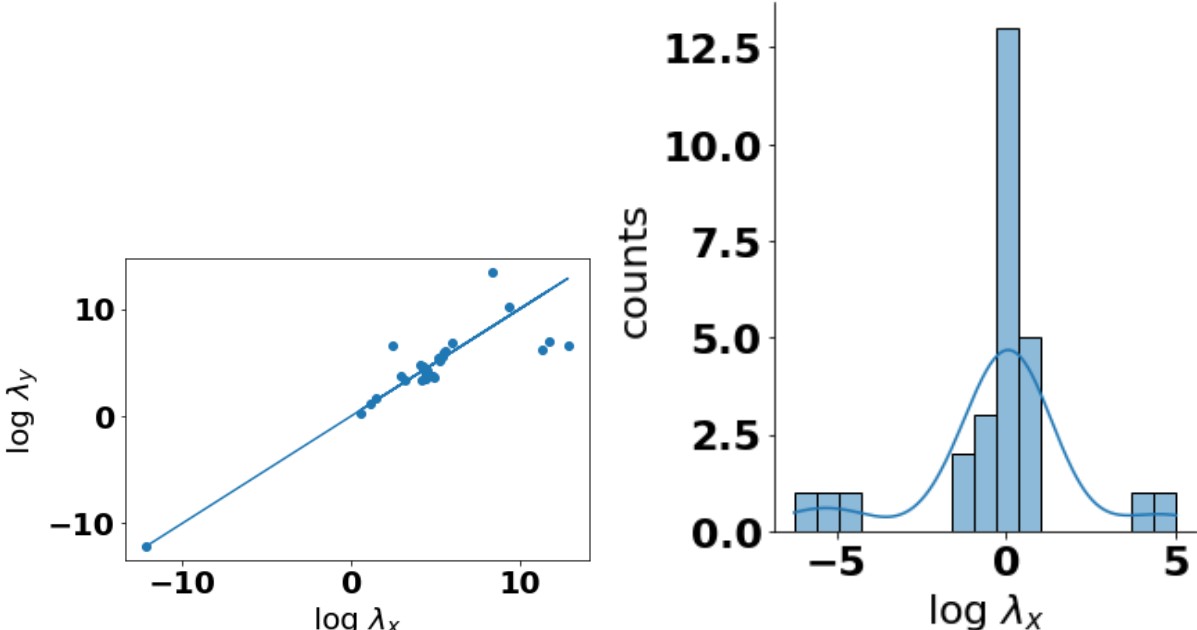

**Figure 10.** Scatter plot of both main horizontal length scales determined for aquifers using a Stable model function (left) a kernel-density
estimate of the residuals around the diagonal.

Let us start with anisotropy in the horizontal direction. Our results showed a strong linear relationship between the estimated
log length scales in both directions (labeled $\lambda_x$ and $\lambda_y$ in Figure 10 left). The scatter is centered around the diagonal line, which
is to be expected since the $x$ and $y$ directions are arbitrarily chosen and do not reflect any geological properties that could induce
a meaningful difference between the two. Using the same procedure as above, we can also estimate the distribution around that
center diagonal (see Figure 10 right). In general, this estimate is based on significantly fewer data points ($n = 27$ in case of the
Stable model) and is therefore less reliable compared to the density estimates presented above. As a result a parametric model
should be used to estimate the prior uncertainty, by following the above procedure. Given the tailing indicated in Figure 10,
the short-tailed Gaussian distribution, used in above example, may not be an appropriate parametric model for this situation.
Instead, the use of a long-tailed distribution like the $t$-distribution would be advisable.





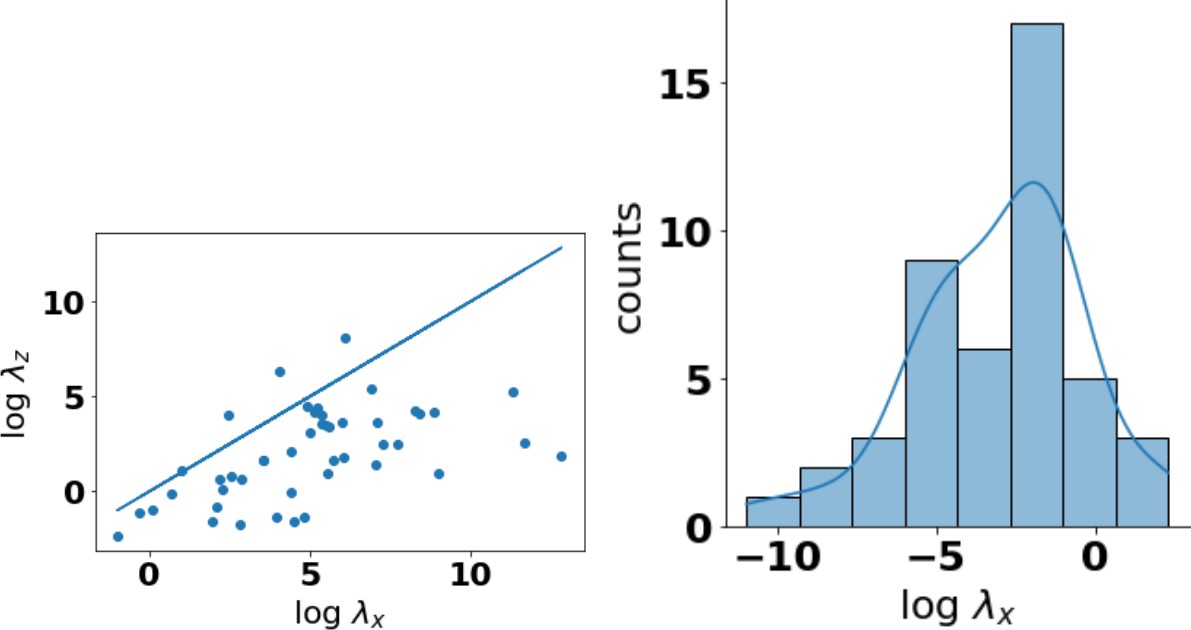

**Figure 11.** Scatter plot of horizontal and vertical log length scales determined for aquifers using a Stable model function (left) and a kernel-density estimate of the residuals around the diagonal (right).

Let us now look at the anisotropy between the vertical and horizontal directions. This anisotropy is known to be strongly
pronounced due to the geological processes like sedimentation (Pyrcz and Deutsch, 2014). Our results confirm this anisotropy
with some notable exceptions (see Figure 11 left). Overall, the number of sites used for this estimation was larger compared
to the above case of horizontal anisotropy($n = 48$ for the case of the Stable model). This number represents only sites in
aquifers but could be increased if sites from soil variograms would be included. Since their numbers are overall small $n = 4$,
we performed no dedicated analysis for this group alone.

One of the most surprising results was the number of cases where the estimated vertical length scale is larger than the
estimated horizontal length scale (see Figure 11 right). They are almost all caused by sites where the estimated length scale
was larger than the maximum length scale. This indicates that it may be, at least in part, caused by the resulting uncertainty
in the estimation procedure. It is consequently not clear whether these results should be used for the derivation of a prior
distribution. If they were to be included, the resulting distribution shows again a long-tailed bell curve behavior. Like in the
case of the horizontal anisotropy, a parametric fitting procedure using the t-distribution could be a good candidate (see Figure
11 right).





### 3.4 Nugget value

The next variogram parameter we investigated was the nugget parameter. This parameter describes the variance at the lag value of zero, i.e., how much do measurements differ that are taken at effectively the same location. Such differences are often

335 interpreted to represent either measurement errors or unresolved variations in the measured variable below the measurement scale (Rubin, 2003; Kitanidis, 2008).

To investigate the behavior of the nugget parameter, we estimated its value using the Stable model function and fitted it against our collected data set. For the analysis, we normalized the value of the nugget against the variance, making sure its value was between $0$ and $1$.

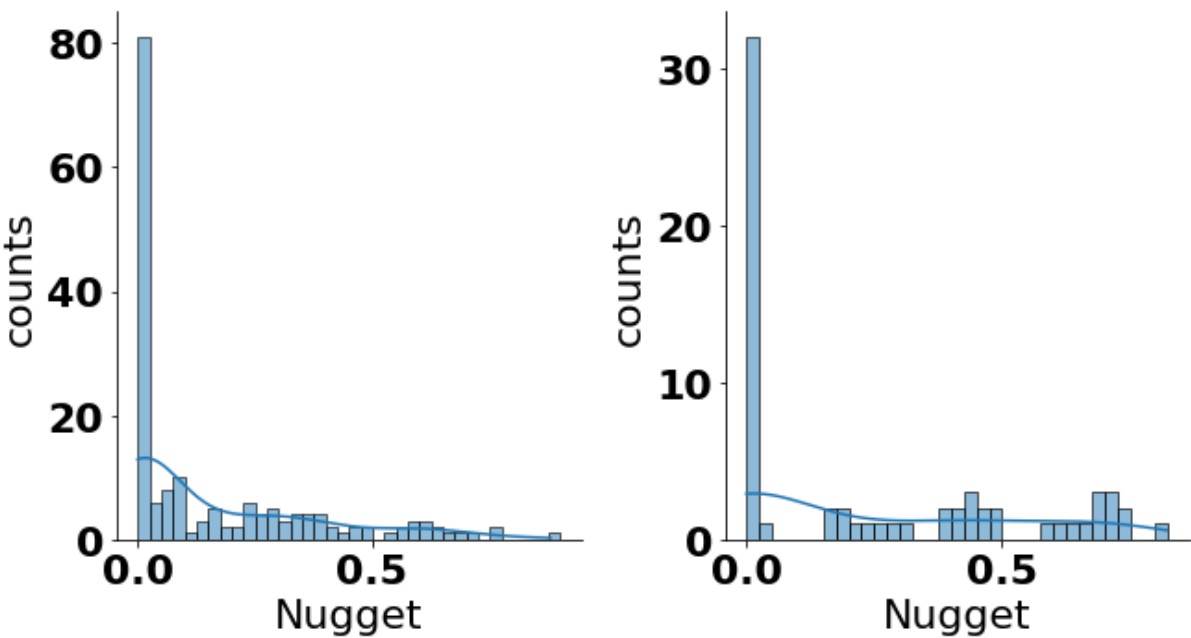

**Figure 12.** Kernel-density estimate of the estimated nugget values for aquifer (left) and soil (right) sites. The used variogram model was the Stable model.

340 Results showed a somewhat similar behavior for the estimated distribution of nugget values for both aquifer and soil sites. In general, most nugget values were close to $0$ in both cases indicating a small or negligible measurement error or sub-scale variabilities. Regardless, a substantial portion of the estimated nugget values were found above the value of $0.5$ meaning that large uncertainties are present in many data sets. Such higher values for the nugget were more common for data sets from soil sites leading to an effectively bi-model behavior of the resulting density estimates. It should be noted that our soil data set was

345 smaller compared to the aquifer data set ($n = 71$ and $n = 215$ for soil and aquifer sites, respectively). As regards a suitable parametric model for this observed behavior, it is clear that a Gaussian or t-distribution aren't viable candidates, due to the potential range of values being bounded between $0$ and $1$. Any parametric model function that is to be fitted against the sample



should, therefore, be chosen to honor both these boundaries as well as the general behavior indicated in the kernel-density estimate. Given the observed behavior in Figure 12, a mixture model using the Beta distribution or a truncated log-normal distribution may be viable candidates.

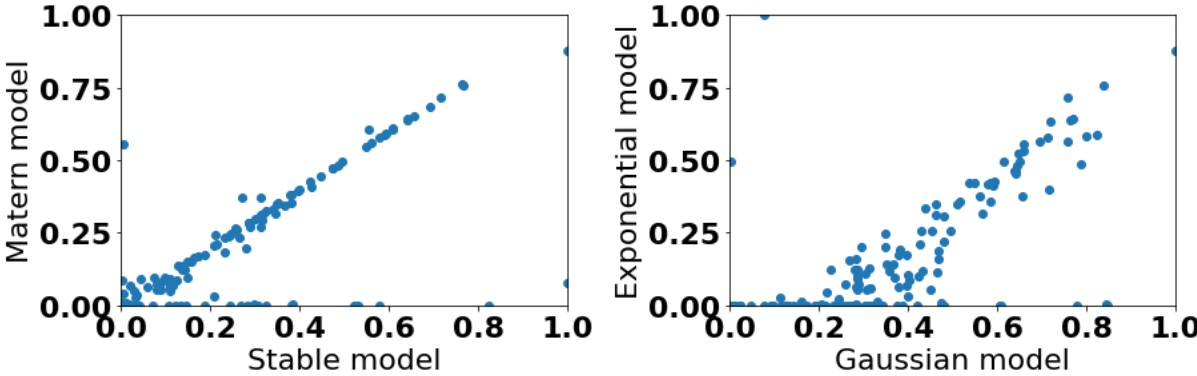

**Figure 13.** Scatter plot of the estimated nugget values for the Stable model vs. the Matérn model (left) and Gaussian model vs. the Exponential model (right). All data were drawn from aquifer sites.

To investigate how the nugget value differed between the variogram model functions, we also compared their respective estimates. Our results showed a strong linear correlation between the estimated nugget of the Stable model and the Matérn model (see Figure 13 left), which shows the similarity between both model functions. On the other hand, plotting the estimated nugget of the Gaussian model vs. the Exponential model shows substantially larger differences between the two (see Figure 13 right). This is due to the different behavior of these two models for small lag values. Whereas the Exponential model exhibits a steep gradient, the Gaussian model is essentially flat in this region. The different nugget values are therefore an artifact of the fitting procedure which tries to compensate for this difference through adjusting the nugget value. This demonstrates that prior distributions for this value should be considered as model specific and should not simply be transferred between different model functions.

Estimated nugget values of the Spherical model showed the highest correlation with the nugget values of the Exponential model but lower correlation with nugget values of all other investigated variogram model functions (data not shown). This is due to the similar behavior of the Spherical model and Exponential model at small lags showing again the relationship between this near-origin behavior of the model function and the ability of the nugget to compensate for any possible mismatch between the empirical variogram and the behavior of the model function. Although all shown results were derived using aquifer sites only, using data from soil sites supports these statements, too.





## 3.5 Shape parameter

Many common variogram model functions like the Exponential and the Gaussian model are fully defined by specifying the length scale, the variance and the nugget value. There is, however, a class of variogram model functions that feature an additional degree of freedom. In the following, we will call this additional parameter the shape parameter.

In case of the well-known Matérn function, this value is known as the roughness parameter $\nu$. This name refers to the fact that its value is directly related to the roughness of the resulting spatial random field (Banerjee and Gelfand, 2003; Diggle and Ribeiro, 2007). This relationship is such that a low value means a high roughness, with the value of $\nu = 1.0$ resulting in a random field that has no derivatives whatsoever, i.e., infinite roughness. A Matérn model function with such a low value is mathematically identical to the Exponential model function. On the other end of this spectrum, a very high value of $\nu \to \infty$

results in a field with an infinite number of derivatives, i.e., infinite smoothness. A Matérn model function with such a high value of $\nu$ is mathematically identical to the Gaussian model function.

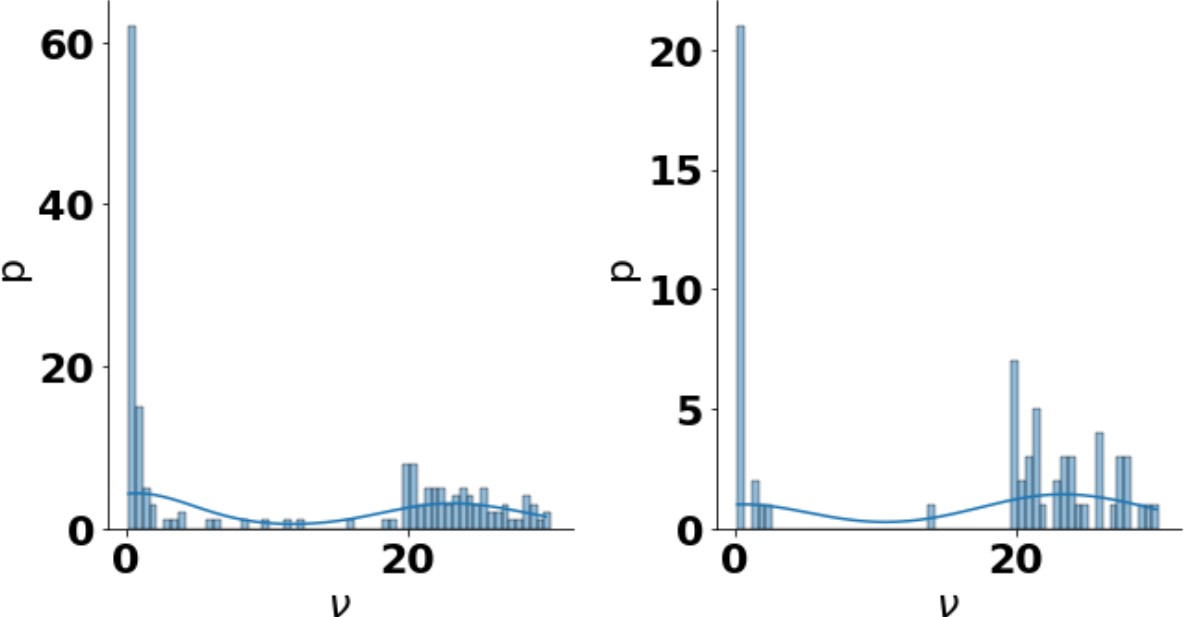

**Figure 14.** Kernel-density estimate of the estimated roughness parameter $\nu$ of the Matérn variogram model function for aquifer (left) and soil (right) sites.

     Our results show a somewhat bi-modal behavior of the resulting frequency distribution of estimated $\nu$ values (see Figure 14). This behavior is very similar for both aquifer and soil sites. The first cluster of the estimated shape parameter $\nu$ is found for very small values with most values being at or near $\nu = 0.5$. This indicates that an exponential model function would

perform with similar accuracy in these cases. On the other hand, a second cluster cluster can be found for $\nu > 20$. Although the Matérn function only converges to the Gaussian function in the limit of $\nu \to \infty$, it should be noted that already for values





of $\nu > 10$, both functions become virtually indistinguishable. The roughness parameter simply loses its sensitivity for higher values meaning that it barely changes the behavior of the function anymore. This means that a Gaussian model function would be able to similarly describe cases in this second cluster very well.

These observations support a number of conclusions. First, despite the roughness parameter spanning a significant range, most of its values fall into two intervals, both of which can be approximated well with a more common model function, namely the Exponential and Gaussian function. Second, the number of cases where a Gaussian model function would be a good fit is larger than expected. Due to its high smoothness, the Gaussian model function is sometimes considered unrealistic (Stein, 1999). This assessment is not supported by our findings at least not from a simple fitting perspective. Finally, the Matérn model

function is still a relevant model function since it may not be clear in advance which classic function, i.e., the Gaussian or the Exponential, can provide a better performance.

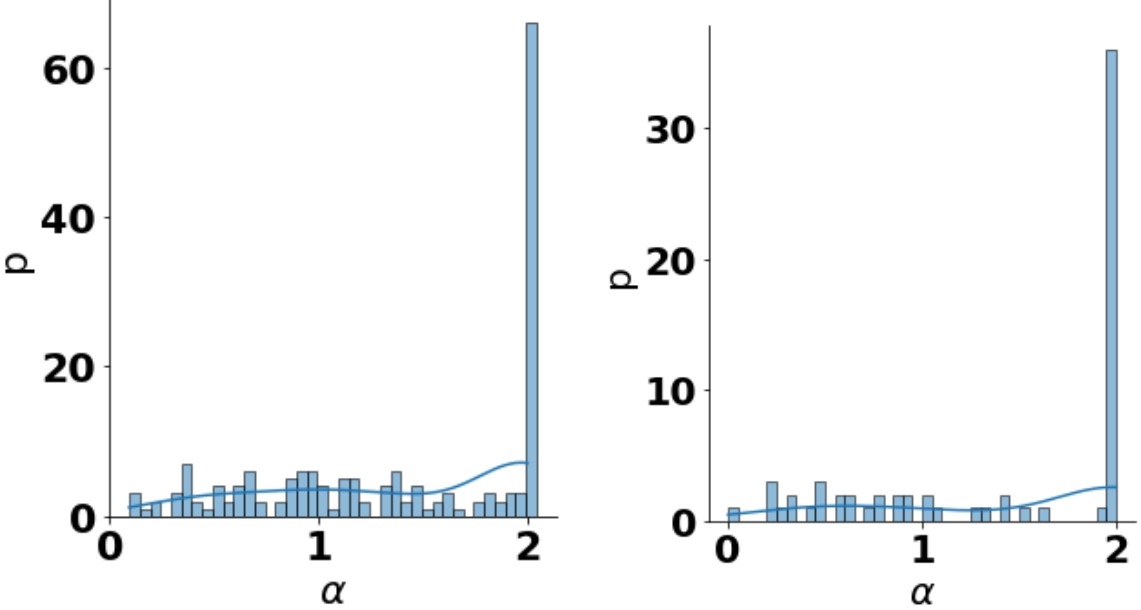

**Figure 15.** Kernel-density estimate of the estimated shape parameter $\alpha$ of the Stable variogram model function for aquifer (left) and soil (right) sites.

In the next step, we analyzed our data set using the Stable variogram model function. The shape parameter of this model function is noted as $\alpha$. Our results show again a roughly bi-modal behavior of the resulting frequency distribution of $\alpha$ (see Figure 15). It should be noted that the shape parameter $\alpha$ is defined between 0 and 2 and many values are found for $\alpha = 2$.

Still, the overall similarity shows a connection between the two shape parameters of the Matérn and the Stable mode functions.

To better understand this connection between the shape parameter $\nu$ of the Matérn model and the shape parameter $\alpha$ of the Stable model function, we performed a regression analysis for those sites where both model functions did result in a fit. Our results showed a very similar behavior for both aquifer and soil sites (see Figure 16). As can be seen, the scatter plot reveals that





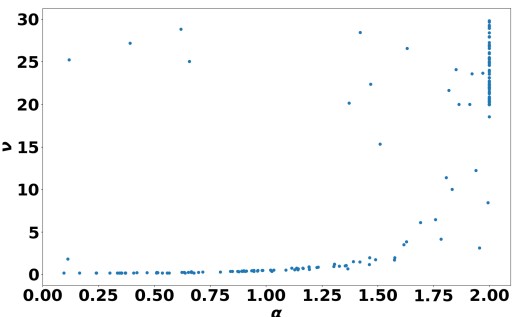 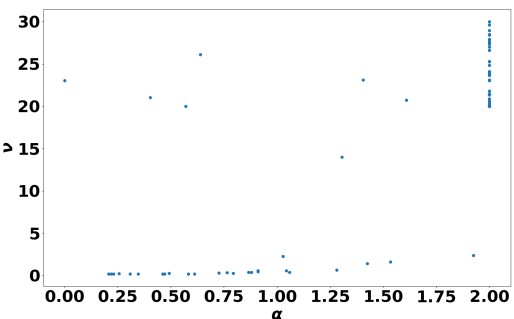

**Figure 16.** Scatter plot of the estimated shape parameter values for aquifer (left) and soil sites (right).

most points in the plot fall into two distinct correlation regimes between $\nu$ and $\alpha$. First, for smaller values of $\nu$, representing an
exponential-like behavior of the Matérn function, we see a linear correlation between the two with a very flat slope. This flat
slope is caused by the strong clustering of sites where $\nu \approx 0.5$ For high values of $\nu$, representing a Gaussian-like behavior of
the Matérn function, we see a nearly vertical behavior, i.e., a larger range of $\nu$ values corresponds now to a very small range of
$\alpha$ values. The latter is caused by the truncation behavior of the Stable model which is confined to values between $0$ and $2$. This
means that the range of $\nu$ values, representing a Gaussian-like behavior, gets mapped into a very small interval of $\alpha$ values
close to $2$ (see behavior in Figure 16).

In general, we can draw two conclusions from these observations. First, the shape parameter $\alpha$ of the Stable model is indeed
related to the shape parameter $\nu$ of the Matérn model since both are directly connected to their respective roughness information
as described above. Second, the confined parameter range of the Stable model is not a drawback from a practical point of view
since the sensitivity of the Matérn model becomes extremely low for larger values of $\nu$. In fact, from a numerical perspective,
this limitation of the parameter range is an asset since it improves the performance of an optimization algorithm necessary
for the fitting procedure. Although this study does not aim to investigate this issue in detail, we did indeed observe a much
higher numerical stability of the Stable model compared to the Matérn model. This stability was observed both in terms of the
number of steps necessary to find an acceptable fit between the model function and the empirical variogram function as well as
in terms of the number of sites for which optimal parameters could be found in the first place. Although the name of this model
is derived from the Stable distribution (Wackernagel, 2003), it, therefore, also describes its numerical behavior, a connection
which is no doubt a coincidence.

One thing that stood out from the data sets being used from the literature was the general lack of data for short lags. Most
of these data sets were generated from observation networks that followed a regular grid layout. This makes sense since most
studies try to maximize the spatial coverage of their measurement campaign but have only a limited number of observation
points due to budgetary constraints. It is, however, problematic from a variogram estimation procedure. A good compromise
would be to arrange at least some of the observation points in a logarithmic fashion (Müller et al., 2021).





### 3.6 Critical assessment of results

In addition to presenting and discussing our results, we would also like to assess our work critically, in the sense of determining both possible weak points as well as limits to their applicability. This will help practitioners to apply our results and use our
data more appropriately and avoid misuse.

The first topic that we would like to address concerns the problem, of publication bias or survivor bias (Schmitz et al., 2012). This is caused by the fact that our data set is based on published data alone, meaning that only data which both the author(s) and editor(s) deemed suitable for publication could end up in our collection. This notion is for instance substantiated by the fact that all empirical variogram functions we found produced viable variogram parameters when they were analyzed by the respective
author(s) of the study. However, when we re-analyzed them, a certain number of sites resulted in a fit where the estimated length scale was larger than the largest length scale in the study. This indicates that authors choose not to publish results that they deemed unsatisfactorily for one reason or the other. As a result, our data set is not a random sample of aquifer and soil sites from all over the world but skewed toward sites where an acceptable variogram analysis was achieved whereas problematic cases may have been left out. Having a non-random data set is a serious challenge for any statistical investigation. Whether
this is a problem, however, depends on the type of application. For a typical geostatistical characterization of a site, it may not be of relevance. After all, practitioners of subsurface geostatisitcs by definition are only going to use these results for sites which they deem appropriate for a variogram analysis. For such an application, the data sets used for our investigations may, therefore, not be biased in any relevant way. Still, there are applications where the topic of survivor bias should be considered carefully. Any situation where our data set is to be used for inferring general properties of aquifer and soil sites, proper care in
the interpretation of one's results is, therefore, advised.

The next topic concerns the variable number of data points used for the inference of the different density distributions of variogram model parameters. While any density estimation improves with the number of samples being used, there are no widely agreed rules as to how many sample points are necessary for an acceptable estimation procedure (Dell et al., 2002). In addition, different features of a distribution need different number of sample points, with higher moments or higher dimen-
sions needing more data (Silverman, 1986). This is particularly problematic for densities having uncommon features like long tails, being highly skewed or being multi-modal. In general, non-parametric estimators can handle the challenge of uncommon distributions well but require a large sample size. On the other hand, parametric approaches require much less data but can lead to model errors if the parametric model is far from the true density (Li and Racine, 2006). In order to account for this problem, we presented an approach where we started with a non-parametric method, like kernel-density estimation, subsequently inter-
preted the results within the context of a suitable parametric model for the inferred behavior of the underlying distribution and then used said parametric model for estimating the density. Of course, this is only one possible approach to addressing these challenges, and practitioners may find other approaches more appropriate depending on their circumstances. Since all data and analyses are openly available, they can easily adapt this approach to their needs.

Related to this topic is the problem that any inference based on past observation may miss features that are not represented
in the used data set (Billot et al., 2005; Gilboa et al., 2010). For the results presented here, this is not of primary concern since





all investigations were performed with respect to properties which are known to be relevant based on prior experience, i.e., the parameters of widely-used variogram models. Still, the data set collected for this study can be used to investigate a number of questions, some of which have been alluded to above. In these cases, this challenge should be kept in mind.

Another topic that needs to be addressed is the fact that some of the variogram data we used for the analysis were provided in clustered form such that they were labeled as coming from the same site but representing different categories. In the data set, we marked these variograms by using the same `site_id` but distinguished them by the label `geological_unit`. As mentioned in the above methods section, the reason why the same site label was used for different data sets differed in each situation. In some cases, the authors separated the data according to different geological layers; in some cases the separation represented geological subunits, subdivided by the authors according to their domain knowledge; in some cases it represented several actually different sites that were combined into a single measurement campaign; and in some cases it was not clear according to which criterion the separation was made. The label `geological_unit` does therefore represent a number of different and disparate situations. Still, the sheer fact that they may be similar can pose a problem from a statistical point of view since variograms from the same site, regardless of what that term meant in that particular study, may be correlated to a certain extent. This problem is known as *pseudoreplication* in the literature (Hurlbert, 1984). Using only a single data set per study would avoid this problem but reduce the overall amount of data available. On the other hand, using all the data risks giving too much weight to some sites, where several variograms are available. To determine the relevance of this risk, we looked at variograms derived from different subunits and saw moderate correlations in some cases and none in others. Within the scope of this study, we did therefore consider these different subunits as independent data points. To properly account for the possibility of within-site correlations, however, a hierarchical model could be employed (Cucchi et al., 2019). In such a hierarchical model such within-site correlation could be estimated from the data provided enough data points are available. Within the scope of this study, we did not perform such an investigation, but the availability of the data set, where variogram data from the same site are marked as such, makes it possible for future investigations to address this topic, if necessary.

The last topic we should discuss is the fact that the empirical variogram functions do not represent raw data but are already processed to a certain degree. This means that these data implicitly contain modelling assumptions that were used when these empirical variograms were determined and are no longer present. As a result, it makes them somewhat less comparable. From a Bayesian point of view this means that the density estimates contain modelling uncertainty, which may, depending on the need of the practitioner, result in a larger uncertainty. This issue is unproblematic from a cautionary point of view, since the result is simply an increase in uncertainty. On the other hand, it is unsatisfactory due to said increase in uncertainty, which means a loss of information, compared to the use of the raw data instead. For instance, for the results presented here, we did not use tertiary data of site statistics due to the modeling uncertainty associated with them. While primary data have the lowest modeling uncertainty, their overall numbers were too small. As a result, secondary data formed the majority of the data providing a compromise between sample size and accuracy.





## 4 Conclusions

In this study, we have presented two different advances for the field of subsurface geostatistics. First, a data set of empirical
variogram functions from a variety of different locations around the world. Second, a series of geostatistical analyses aimed
at examining some of the statistical properties of such variogram functions and their relationship to a number of widely used
variogram model functions.

The data set collected for this study is freely available at the online repository associated with this manuscript (see below
Section 'Data and code availability'). It can therefore be used by practitioners to replicate our analyses, extend it with additional
data, and adapt them to their needs. They can also use it to explore new questions not covered here. Finally, we explicitly
encourage practitioners to both expand the data set and extend the range of meta information associated with it. This would
allow to answer additional questions and broaden the scope of the data presented here.

As regards our analyses of said data set, we have derived a set of frequency distributions for the parameters of variogram
models that can be used as prior distributions for Bayesian geostatistical applications. Since these prior distributions already
contain a considerable amount of information, their use will result in a higher information content in the posterior. Given
the overall dearth of subsurface data and the often high exploration cost, such an additional source of information presents a
valuable asset for a geostatistical characterization of a soil or aquifer site.

In addition, we investigated the viability of different variogram model functions for modelling the empirical variogram cloud.
Our results showed an overall similar accuracy of all investigated variogram model even though some feature one additional
degree of freedom. This overall similar accuracy supports the notion that variogram models can be primarily chosen by the
practitioner based on other considerations like familiarity, applicability and availability.

Finally, our investigation revealed the distribution of some geostatitical features of subsurface sites. First, the widely-
observed scale effect of many subsurface properties is strongly pronounced for the characteristic length scale of the hetero-
geneities. This observation supports the conceptualization of the subsurface as a fractal medium, where heterogeneities appear
on any scale of observation and their apparently finite length is, at least in part, a finite-size effect caused by the truncation of
the measurement process. Next, the nugget value, a feature representing measurement errors and sub-scale variability is widely
distribution over its possible range, an observation that is exacerbated by the fact that simpler variogram model may tend to
compensate with the nugget parameter for a mis-matched model behavior at short distances. Finally, that behavior at short
distances is strongly connected to the roughness of said heteroneities. Our results show that most sites fall into two distinct
categories depending on that roughness, i.e., either having very high or very low roughness. If this behavior is to be represented
correctly, a more flexible model function, e.g., the Matérn or Stable model, is to be used.

To extend the results and data discussed here, a number of options can be considered. First, expanding the number of
sites covered and adding more features could reduce the uncertainty in the prior distributions. Using the above workflow,
the uncertainty in these distributions represents the uncertainty of the entire data set and thus assumes that a particular site
is a random draw from that set. However, it is not mandatory to use such a large and therefore statistically highly variable
population. In fact, there is no unique population from which any given site needs to be considered to be random draw from;



a notion that is known in statistics as the *reference class problem* (Hajek, 2007; Hajek and Hitchcock, 2016). As a result, it is advantageous to use the most precise reference class for which a large enough sample is still available, thus striking a balance between precision and accuracy (Wallmann, 2017). In subsurface geostatistics, this would mean to use only sites for the transfer

of information which are similar to the given site based on some criterion of site similarity (Kawa et al., 2022). Yet, being able to limit one's analysis to a smaller, more appropriate, and less variable cluster of similar sites would require a large population of sites, arguably larger than the current data set.

Another possible venue for further study could be to establish a connection between certain variogram properties and geological features of the site. This would again necessitate the addition of geological features to the data base itself, a task that

was beyond the scope of the current study. If done, it could, e.g., help practitioners to discern the viability of a given variogram model or of a variogram-based modeling approach in the first place.

*Code and data availability.* In this study, we used a number of software packages for the preparation of the data and the analysis of the results. To guarantee that others can make use of the data collected in this project and reproduce and adapt our analyses, we provide online resources to make them available. They are as follows:

– For the variogram/covariance analysis, we used the `GSTools` Python package (Müller et al., 2022). This software is developed at https://github.com/GeoStat-Framework/GSTools. The used software version was 1.3.1 (Müller and Schüler, 2021).

  – The data used for the analysis in this manuscript is provided at the https://github.com/GeoStat-Examples/GeoStat-DB GitHub repository inside the `data_raw/`, `data_prep/`, `data_proc/`, and `data_stats/` folders. Since this online version will be updated constantly, we also created an Zenodo repository for the data used exclusively for this manuscript (Heße, 2022).

– The workflow to reproduce the analyses from this paper and the figures used herein is provided at again at GeoStat-Examples/ GeoStat-DB in the `src/` folder.

*Author contributions.* Falk Heße is the main author of the study. As such, he conceived of the design, collected the data, implemented the workflow, analyzed the results, created the figures and contributed to all parts of the manuscript. Sebastian Müller is the main creator and maintainer of the the `GSTools` package. As such, he provided the necessary help for the setup of the numerical workflow, helped with the

interpretation of the variogram analyses and contributed to the methods section of the manuscript. Sabine Attinger provided her assistance and expertise during the study as well as for the completion of the manuscript.

*Competing interests.* The authors declare that the research was conducted in the absence of any commercial or financial relationships that could be construed as a potential conflict of interest.





*Acknowledgements.* For this work, Falk Heße was financially supported by the Deutsche Forschungsgemeinschaft (DFG) via grant no: HE
550   7028/2-2.



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
