# Peer review of "Data-driven estimates for the geostatistical characterization of subsurface hydraulic properties"

_Hydrology and Earth System Sciences, 2023_

## Author Response (AR1)

**reply:hess-2023-15-MS**

July 2023

Dear Jesus,

thank you again for editing our manuscript. Please find attached the revised version of the manuscript, as well as a version of the manuscript with the changes highlighted and our responses to the three reviewers. In these responses, we have presented the reviewers' comments in black font, followed point-by-point by our response in blue font. We hope that the revisions to our manuscript and our responses address your and the reviewers' concerns and look forward to hearing from you.

Yours, Falk

**reply:hess-2023-15-MS**

May 2023

**1 First reviewer**

The paper "Data-driven estimates for geostatistical characterization of subsurface hydraulic properties" by Hesse et al. is a nice try to build a comprehensive data set of variogram parameters that could be used as reference values or as prior distributions for cases in which there are not enough data to construct an experimental variogram from the available observations. While the proposal seems interesting, I have many reservations about the paper's contents and how the results are obtained and analyzed.

The reviewer presents long and thorough discussion of our manuscript, which raises a number of points considered critical. At least some of these more critical are, in our opinion, caused by a misunderstanding of our aims and the methodology employed here. To avoid such a misunderstanding in the future, we have revised our manuscript and now explain the points more accurately. Others are important and we hope we have addressed them in the revised version of the manuscript. In the following, we will list the reviewer's comments and present a point-by-point reply, where we address them and show how we revised the manuscript to account for them.

First of all, pretending to find the "universal" variogram parameters distribution is a little bit pretentious. The analysis is made from a purely mathematical/computer science perspective without considering that variograms relate to the underlying geology. In this sense, one would not expect variograms computed in fractured media to be the same as those computed in sedimentary formations. Also, and very importantly, a parameter such as hydraulic conductivity is not expected to vary in space smoothly under any circumstances; for this reason, the use of Gaussian variograms or, in general, variograms with low roughness should never be advocated (unless they are accompanied by a nugget effect that would break the smoothness imposed by those variograms).

This comment presents several of the misunderstandings in our opinion. First, at no point do we use the term "universal" for the derivation of our variogram parameters. Quite the contrary! We make it clear numerous times throughout the manuscript that the parameters derived in our study strongly depend on both the model and the data being used to derive them. Second, of course the estimate would change to some degree if the data set would be split

into, say, fractured and porous media. We did, for example, recently publish a manuscript where we investigate the impact of such a re-clustering of the data on a Bayesian estimation procedure [Kawa et al., 2022]. However, in this study, we determined that with the given sample size, only a modest gain in uncertainty reduction was achieved. We therefore, argued in the manuscript, among other things, for an increase of available data for subsurface characterization. Since the sample size used in this study was somewhat smaller than the data set used in the aforementioned study, we did not attempt to split the sample even further. We discuss this problem of accuracy vs. precision, which affects every statistical analysis in the discussion. Thirdly, we do explicitly not advocate for the use of the Gaussian model function. In fact, we do not advocate for the use of any model function in particular. The choice of variogram model functions in our study was only motivated by the fact that they are regularly employed. We do not, can not and don't want to make any comment on whether their use is justified or not. If the reviewer wants to argue in favor or against the inclusion of any model function based on the above criteria, we would happily include or exclude any model function. In fact, following the comment of two other reviewers, we did include and re-did the analysis with respect to one other model function, namely the Truncated Power Law variogram. We also reduced the use of the Gaussian model function. For instance, in the example on how to derive a prior distribution for the characteristic length scale, based on the maximum length scale, we now use the Exponential model function.

Next, the terminology used departs sufficiently from the one used in standard geostatistical literature (see, Journel and Huijbrets, 1978; Isaaks and Srivastava, 1989; Deutsch and Journel, 1991; or Goovaerts, 1997) to make it very confusing, and, at some points incorrect.

The terminology we follow in our manuscript follows the notation we used in Müller et al. [2022]. This notation in turn is a compromise between different notations used in the wider geostatistical community since this is the target audience for this Python package. We agree, that some notation decisions could be justified more and we will update the manuscript accordingly. Further details will follow in the responses below.

Let's dive into more detail about my reservations. The variogram is a vectorial function that depends on the separation distance vector h; only in the case of isotropic variograms in all directions does such a function becomes a scalar one. The definition in section 2.2 and the analysis thereof is made assuming that the variogram is a scalar function depending only on the modulus of h. This is a significant conceptual flaw, which is not cleaned with the anisotropy analysis made in section 3.3. The expression of the variogram function in line 135 is incorrect. You cannot separate the variance and the nugget effect. The variance of the random function model is (n+sigma2) (using the notation in that section), and the covariance is the variance minus de variogram ((n+sigma2)-gamma). The authors should recall that, generally, the variogram is modeled as a sum of variogram functions, each one contributing their share to the total variance (or variogram sill). They have chosen to model it as the sum of two variogram models, a nugget effect (with a contribution of n) and another parametric function (with a contribution of sigma2), which is perfectly OK; however, the authors should acknowledge that there could be fitting with more than two components. What is not OK is much of the later analysis in which this fact is disregarded: many of the findings are due to this fact, the fitting of a nugget plus another variogram, and some of the conclusions are said to be due to, say, the type of parametric model used, disregarding that a nugget had been used to fit the variogram.

We choose the simple representation of the variograms for the sake of simplicity of the formulas within the paper. Within GSTools we of course provide spatial variograms as a function depending on a vector.

The details of the specific variogram formulation are explained in detail in Mueller et al. (2022) (paper about GSTools v1.3).

Anisotropy is also explained there with the use of an "isotropic distance" that is a scalar value where the distances along the the main axis of correlation are incorporated and re-scaled with the respective ansiotropy ratios. We could add a short section referring to this to clarify the assumptions.

Very important, sigma2 is not the variance of the process! It is only the contribution of one of the two structures fitted to the total variance.

That is of course true. We use the term variance in this context for the correlated variability (or partial-sill), meaning the portion of the sill above the nugget ("sub-scale variance" or "uncorrelated variability"). The sill or total variance is then the sum of these two. The nugget was incorporated into all models to not create a special case to always add a pure nugget model to an existing model just to get this feature. We now better explain this important distinction in the revised version of the manuscript.

The variogram computed as a function of the modulus of the separation vector is referred to as the omnidirectional variogram. It is generally used during the modeling process to assess the nugget effect and the sill of the variogram and to have an idea of the average range from the different directions, but it is not a variogram to be used in practice for anisotropic media. Variograms are vectorial functions, and as such, their value depends on the vector orientation reflecting the anisotropy in the correlation patterns of the variable under analysis. Such anisotropy is characterized by an ellipsoid (in 3D) or an ellipse (in 2D), which, in principle, may be arbitrarily oriented, although, in practice, it is assumed that one of the ellipsoid axes is oriented vertically and the other two in the XY plane, not necessarily parallel to the Cartesian axes. The lengths of the semiaxes of this ellipsoid are known as the anisotropy ranges and are crucial to characterizing the spatial continuity of the attribute. The authors disregard the vectorial nature of the variogram and introduce the "characteristic length" as one of the parameters of the parametric variogram components. The characteristic length, 'l' in the equations in the lines between 135 and 150, is not the range and may induce mistakes. In standard geostatistics, the range is defined as the distance at which the variogram value reaches the sill, or for variograms that are asymptotic to the sill, the distance at which the variogram reaches at least 95 % of the sill. This means the range will equal l for the spherical variogram but 3l for the Gaussian or exponential ones. Comparing the value of l for different variogram functions

is like comparing two different quantities.

We wanted to keep the section about the theory and assumptions behind the used variogram formulation quite brief but we agree that certain aspects could be made more clear. First, we are mostly following the formulations of Rubin [2003] where a "non-dimensional" distance is used in the variogram formulation to reduce the spatial variogram to the omnidirectional one. For this, the lag distances along the main axes of correlation are rescaled with the respective anisotropy ratios and the axes are rotated to follow the principle axes. This was explained in detail in the paper about GSTools [Müller et al., 2022]. We will add some more explanation about this in the manuscript. Second, the criticism on the term "variance" is understandable and we will substitute the term "variance" with "correlated variability" as used by Rubin [2003] (see our answer above). There, this variability is also denoted with $\sigma^2$. The reviewer is right, that the total variance (and/or the sill of the variogram) $s$ is then the sum of the sub-scale variability $(n)$ and the correlated variability $(\sigma^2)$: $s = \sigma^2 + n$. GSTools includes the nugget in all models as a separate parameter since the pure nugget model would be the only one with a totally different parameter set. Cressie [1993] also includes the nugget in the model examples. Third, GSTools implements anisotropy by using rotation angles and anisotropy ratios that unify the variogram formulation with the omni-directional scalar variogram function by using the non-dimensional distance $h$ as just described. We will add a paragraph clarifying this. We also do agree with the reviewer that comparing parameters between different model functions can be very elusive due to conceptual differences between these functions and the role these parameters play. This is why we constantly warn against the simple transfer of results derived for one model function to another and make it clear that every statement we make is contingent on the function being used.

I am very concerned with the many published theoretical papers in which the variogram is assumed to be Gaussian. This assumption is always a matter of convenience, especially in those papers more focused on analytical approaches, disregarding the fact that Gaussian variograms make only sense for variables that vary smoothly in space, such as the thickness of a layer, but never should be used to model a parameter characterized by a very large spread with significant short-scale variability like the hydraulic conductivity.

We are aware of the long standing debate regarding the Gaussian model function and the high smoothness it introduces into the modelling process. We explicitly mentions this topic and how it may be detrimental to the applicability of said model function. However, our study is not intended to make any statement regarding this issue. The only criterion for inclusion was whether the model was regularly used. This is certainly the case for the Gaussian model, however justified one may think this is. As such, we do not consider this inclusion of this model function to be an endorsement in any way. However, to reduce the use of the Gaussian model function we changed the example of the derivation of the prior distribution. Instead of the Gaussian model function, we now use an Exponential model function in the revised version of the manuscript.

It is not clear if the variograms analyzed are for K or logK. Line 157 mentions

K, and nowhere is logK mentioned. The variogram of K may not of interest in geostatistical studies since it is common to use a lognormal random function to model the spatial variability of K. In addition, if K had been used, my criticism of using Gaussian variograms (or variograms with low roughness) is exacerbated.

This is an important observation, which was also criticized by another reviewer. Of course, all the analysis refers to the log-hydraulic conductivity. To better convey this important fact, we revised the manuscript accordingly.

Considering the significant uncertainties associated with experimental variograms, trying to distinguish whether a stable or a Matern variogram fits better than an exponential or a spherical one is meaningless. The analysis would have benefited from comparing just the spherical and exponential variograms (plus the nugget effect). Neither the Matern nor the stable models make sense when modeling hydraulic conductivity from a geological point of view. A much more meaningful fitting would have been a power variogram, which would relate to the fractal behavior claimed by Neuman.

Following the similar comments of the two other reviewers, one of them being Shlomo Neuman, we re-did the analysis by including the a fractal model function.

More specific comments

What are observable base rates? (line 22)

Observable base rates refers to the notion that informative prior distributions in Bayesian inference should be based on observable frequencies. To avoid this misunderstanding, we removed the term from the manuscript.

What is a variogram cloud? (line 39, line 96, and many more times in the following pages)

This term refers to the empirical variogram function. We added an explanation to the manuscript to make this more clear.

The paragraph from lines 52 to 55 is meaningless.

We do not agree. In fact, one other reviewer encourage us to make the comparison of established variogram models even more explicit.

After introducing the roughness coefficient in line 166, the authors analyze the roughness of the different parametric functions used in their variogram definitions, but they fail to recognize that as soon as a nugget effect is added, the roughness of the fitted variogram is zero. This explains why, later, some of the analyses seem to favor fitting a "Gaussian variogram" when in truth, they are fitting a variogram composed of a nugget effect plus a Gaussian component. The nugget removes the implicit roughness of the Gaussian variogram, justifying its application in practice.

This is a very important observation. What we saw in our study was that the Gaussian model can describe very well empirical variogram clouds. So from a pure fitting perspective, its use seems justified. We did not further comment of this but simple left the observation as is. The observation by the reviewer could be an explanation as to why the model seems to perform so well in this regard, despite the shortcoming often discussed in the literature. We threfore revised our discussion section and added this observation to the manuscript.

When introducing section 3, the authors discuss the length scale and vertical and horizontal anisotropy. For the first time, the authors recognize that anisotropy may be an issue, but it is poorly defined. The length scale could be the larger range, and the horizontal and vertical anisotropies could be the ratios of the ranges in the other two principal directions with respect to the largest range. But this is not what the authors analyze; they focus on the range of the omnidirectional variogram, plus the ranges in some arbitrary directions in the XY plane (not necessarily in the directions of the principal ranges) and the range in the vertical direction.

As said in the manuscript, the majority of data that analyzed anisotropy was secondary data, ie., the authors already did separate the data according to this criterion. However, the anisotropy for the horizontal directions is one result where primary data form a substantial amount of the overall data. We, therefore, re-did this analysis. In general, what we saw from secondary data, the practice was to align the x- and y-direction with the physical directions indicated by the measurement campaign and not be some possible anisotropy observed in the data. This may not be the best way to perform this analysis, but we had to work with the data we have, not with the data we wished for. As described above, we did analyze the directional variogram and not only the omni-directional variogram. The assumptions and implementations details for this are discussed in Mueller et al. (2022) (paper about GSTools v1.3). This is now better explained in the revised version of the manuscript.

In section 2.3, it would be nice to discuss the fitting procedure used to get the variogram parameters.

For all the analysis, we used the tools provided by GSTools. The estimation of variogram parameters is, for instance explained in Müller et al. [2022] and one the website of the project. We are not sure what aspect of the fitting procedure should be added.

In the fitting algorithm, the authors should use an algorithm capable of fitting a fully vectorial function and attempt to identify not just a single omnidirectional range but actually the whole anisotropy ellipsoid (or ellipse), that is, the three (or two) principal ranges and their orientations.

We have to emphasize again that the majority of the data was secondary data. In no case, where primary data was available did we simply fit an omnidirectional variogram. We did acknowledge the shortcomings of the used data in this regard and its ramifications on the topic of Bayesian inference, namely the fact that it increases the uncertainty in the prior distribution. However, the point raised here is important and we, therfore, rephrased our explanation in the manuscript.

The statement in line 215 does not seem to be much justified. Why should all models fit data? There may be physical reasons why one kind of variogram is more appropriate than other. In some cases, as already mentioned, some variogram models are inconsistent with the sample data (such as the Gaussian variogram without nugget effect to fit the spatial variability of hydraulic conductivity).

Of course, not all models should be better at any given data set. However,

this was only a statement in the aggregate. We revised the statement to the manuscript to better explain this notion.

Line 223. The authors realize that the nugget effect is responsible for the similar results yielded but the different parametric models.

We are not sure what this statement means.

The matrices of scatterplots must be explained. What does each point in each scatterplot represent? The diagonal scatterplots make no sense unless they represent something different from the rest of the scatterplots. If they show the same parameters, all points in the diagonal scatterplots should fall in the diagonal at 45 degrees.

We removed one of the matrix scatterplots and added a table instead, to better convey the relevant information. We also redid the other matrix scatterplot to improve the presentation and revised the explanatory statement to provide context to the results

The end of the sentence in line 229 is very important. It is always so! The particularities of the site under analysis should drive the analysis. Trying to find the de "universal" parameter distribution is a mistake.

As we mentioned above, at no point do we attempt to find a universal law underlying all aquifer and/or soil sites. This is a crucial misunderstanding of our aims. Our main aim is to gather and subsequently use a large data set to derive informative prior distributions for Bayesian inference. In addition, we try to discern typical behavior of variogram functions, which may be of use for delineating some challenges of properties found in real-world data for geostatistical applications.

The second sentence in the paragraph starting at line 230 is hard to understand.

We revised this sentence in the manuscript.

Unclear what is meant in the sentence that starts in line 251 to the end of the paragraph.

We also revised this sentence in the manuscript.

End of page 12: A power variogram could explain this behavior.

Following this and the other reviewers' comments, we included a power-law variogram into the analysis.

In the paragraph in line 260, the relationship between the lengths should be at the 45-degree line when comparing spherical and Gaussian and at the y=3x for the comparison with the spherical variogram.

This is an interesting point, which we consider in the revised manuscript.

Section 3.3 makes absolutely no sense. Anisotropy is a fact and should be appropriately analyzed and accounted for. Analyzing lambdax and lambday when they do not represent the actual principal directions of the anisotropy ellipse is useless.

We would like to refer to our answers above.

The analysis of lambdaz would make sense if compared with the largest range in the plane but not when compared with an arbitrary range calculated in the x direction.

Likewise, we would like to refer to our answers above.

In section 3.4, I hope that the normalization of the nugget was done with respect to the total variance (n+sigma2), not with regard to sigma2. There is no reason why n could not be larger than sigma2, therefore escaping from the [0,1] limits.

Again, we would like to refer to our answers above.

The nugget not only accounts for measurement errors but also for short-scale variability. The discussion in this section is poor and, at times, flawed.

We mention several times in the manuscript that the nugget represents measurement error and sub-scale variability.

Line 354 is explained because the Gaussian model needs the nugget to fit most experimental variograms, whereas the exponential does not. Consequently, the results are not an artifact of the fitting procedure.

This is an interesting observation by the reviewer. But wouldn't it still mean that it is an artifact by the fitting procedure?

In the opening of section 3.5, the authors forget that the length parameter is not enough; there is an anisotropy ellipse.

See our answer above.

Why does Figure 15 not show a histogram as before?

Figure 15 does show a histogram like the other figures before.

The discussion about survivor bias is very interesting.

We agree.

The quality of the figures must be improved.

We agree. Several of the figures still showed blurring and in some cases, the font needed to be equalized. We, therefore, re-did all the of the figures by changing the font size, adding grid lines and increasing visibility.

I think a reanalysis of the data accounting for the mistakes made in this evaluation, probably reducing the number of variogram functions and avoiding the construction of the prior probability functions for the different models, would be worth publishing.

We hope that our explanations regarding the real or perceived mistakes have been helpful. As regards the number of considered variogram functions, we followed the advice of the two other reviewers and added a truncated power law variogram to the analysis. As a result, the number of variogram functions has unfortunately increased, instead of decreased. As regards, the construction of the prior distribution, we do not agree at all. On the one hand, the data set we collected for this study allows for a wide range on investigations and we would be happy if people would do this. On the other hand, the derivation of the prior distribution is the main application that we were interested in, when initiating this study. For this reason, we consider it the most important one from our point of view. As a result, we will keep it in the manuscript.

**References**

Noel Cressie. *Statistics for Spatial Data*. Wiley Series in Probability and Statistics. John Wiley & Sons, Hoboken, New Jersey, September 1993. ISBN

9780471002550.

Nura Kawa, Karina Cucchi, Yoram Rubin, Sabine Attinger, and Falk Heße. Defining hydrogeological site similarity with hierarchical agglomerative clustering. *Groundwater*, 2022.

S. Müller, L. Schüler, A. Zech, and F. Heße. `GSTools` v1.3: a toolbox for geostatistical modelling in python. *Geoscientific Model Development*, 15(7):3161–3182, 2022. doi: 10.5194/gmd-15-3161-2022. URL `https://gmd.copernicus.org/articles/15/3161/2022/`.

Yoram Rubin. *Applied Stochastic Hydrogeology*. Oxford University Press, USA, 2003.

review:hess-2023-15-MS
2023-05
journal article review
en

**review:hess-2023-15-MS**

May 2023

**1 Second reviewer**

The paper presents present a data-driven approach to a classical problem in subsurface hydrology, the estimation of parameters characterizing the variogram of subsurface properties. The proposed method advocates the use of Bayesian inference to set up a prior distribution for models that describe spatial correlations (covariance or variogram). A remarkable data set is examined, and available data are classifiable into primary (point measurements), secondary (empirical variogram functions), or tertiary (statistical estimates of subsurface properties) data. Data were processed to avoid overlaps and over-representation.

The first available result from the manuscript is the comparison between different variogram model functions, that could be improved in my view (see comment #2). The scale dependency of the hydraulic conductivity is examined next, confirming earlier literature results. An example application of the Bayesian approach to the estimate of correlation length given the maximum length scale is then presented. Other variogram parameters examined are anisotropy, nugget effect, and shape parameter.

The discussion addresses important issues such as the unbiasedness of the data set employed, among others.

The paper looks as a mature contribution; given the topic and the type of paper, I see however some room for further improvement. Results are of interest to the readership of Hydrology and Earth System Sciences. The methods are adequate, the paper subdivision into sections sound, and the figures illustrative. I recommend minor revisions for the reason explained below.

We appreciate the reviewers comments and overall supportive feedback on our study.

The manuscript examines only stationary variograms, I suggest to mention that nonstationary variograms (see, e.g.,Di Federico and Neuman, 1997) were excluded from the analysis. Due to a similar comments of one of the other reviewers (Shlomo Neuman), we revised our analysis and included a truncated power law variogram into the analysis. The results of this analysis and some discussion on it can be found in the revised version of the manuscript.

The comparison among variograms having a different numbers of degrees of freedom (section 3.1) could be rendered more qualitative by model identification criteria (AOC, AIC_c, KIC, ...), incorporating the number of parameters

involved and the principle of parsimony. The same holds probably for other comparisons performed. We agree with the reviewer that the use of a model-selection criterion can formalize the comparison we have done in the manuscript so far. Models with more degrees of freedom have more flexibility and should therefore be better able to capture any observed behavior. In case of variogram models, the Matern, the Stable model, and, depending on the fitting procedure, the truncated power-law model should therefore outperform simpler models, like the Exponential model, in terms of goodness-of-fit. In the current version, we only performed a qualitative analysis by observing the overall similar accuracy and noting how this would not justify the use of a more sophisticated model like the aforementioned Matern and Stable model. Using a criterion like AOC, AIC_c, KIC, etc. would make this analysis more quantitative. In the revised version of the manuscript, we now provide the numerical values of the pseudo R2-scores for the different model functions and show how the more flexible models perform marginally better than the simpler ones. We did, however, not add a further analysis using a model-selection criteria, since all the performance was overall extremely similar, meaning accuracy concerns are not of primary importance for selecting any given variogram model function. We now explain this reasoning in the revised manuscript.

How crucial is the assumption of two independent Gaussian distributions in Section 3.2 ? Could they develop a more general theory without it, maybe subject to other limitations? The assumption of two different Gaussian distribution is not crucial to the approach presented there at all. In fact, the parametric model for the residuals around the regression line was chosen on the spot after visually inspecting them. As we explain in the discussion section, many different parametric models may be possible depending on the situation. In fact, if enough data points are available a completely non-parametric approach is possible as well. To summarize our approach here again, we would describe it as follows. First, the residuals around the regression line are representing the uncertainty one has with respect to the regression model. From a Bayesian perspective, the can be used to estimate a prior probability. We do this by first visually inspecting the results of a kernel-density estimation (KDE), ie., a non-parametric estimation procedure. KDE is a powerful estimator, but it always produces very smooth densities which may bamboozle practitioners into over interpreting its results. To avoid overconfidence, we therefore only use KDE to find a good parametric model that could describe the empirically observed distribution of the residuals. In our case, a mixture mode using two Gaussian distributions seemed like a good choice. When we fitted such a model to the residuals and compared it to the KDE, we saw a excellent overlap. Of course this agreement has to be interpret with care since the KDE is not the ultimate benchmark of truth, for the reasons outline above. But having two different estimation procedure give very similar results certainly adds confidence that the both express some underlying truth. This single example is explicitly presented as a proof of concept for how to use the data provided in our study for the derivation of prior distributions in a Bayesian context. As we state in the manuscript, using other data and/or other variogram functions may lead to a

somewhat different regression analysis with different residuals. In our opinion, there is probably no general theory on what parametric model, if any, to use for the description of the prior distribution. Every case may be different and practitioners are advised to use their judgement in adapting this approach to their situations. In the revised manuscript, we now explain this reasoning in more detail to better convey this important idea.

**review:hess-2023-15-MS**

May 2023

**1 Third reviewer (Shlomo P. Neuman)**

The authors have conducted an interesting statistical analysis of raw variograms (I assume that this is what the authors mean by variogram clouds) of published data from a variety of geographic locations and geologic settings. Though the data are said to include saturated hydraulic conductivities from aquifers and shallow soils, as well as permeabilities and transmissivities, I assume (but the authors need to confirm or otherwise clarify) that the data consist of logarithms of these variables. A strength AND a weakness of the analysis is the lack of clear distinction between varied hydraulic properties (conductivity versus transmissivity) from varied settings (sediments, rocks, porous versus fractured media) measured by varied means (I presume single and multihole pressure tests, perhaps other) on varied scales of measurement. The strength of the approach is that it reveals commonalities among the data (scaling and amenability to analysis by a variety of variogram models); its weakness is lack of clarity about the way hydrogeologic settings, methods and scales of measurements, affect variogram behavior.

We appreciate the reviewers comments and overall supportive feedback on our study. The reviewer is right that we used the logarithm of the conductivity, transmissivity and permeability values. To make this more clear, we explicitly mention this now in the revised version of the manuscript. As regards the lack of clear distinction between varied hydraulic properties, we agree with the reviewer that, as far as the analyses are concerned, this is both a strength and weakness. Any statistical analysis has to find a compromise between sample size and accuracy. Using the whole sample means that no within sample effects can be discovered. On the other hand, split the sample according to different criteria, and the resulting sub-samples may be too small to perform a viable statistical analysis. For our paper, we only distinguished between soil and aquifer site. Splitting for instance the aquifer data further into conductivities, transmissivites and permeabilities, would facilitate the investigation of possible differences between them. Alas, since most of data was drawn from measurements of hydraulic conductivity, the samples for transmissivites and permeabilities would have been very small, thus, making comparisons rather elusive. Still, the data set is openly available and practitioners interested in this topic are free to do such an analysis by themselves. The same problem of

reduced sample size also applies to comparisons between, say, different settings and/or measurement techniques. This was particularly unfortunate, since we were very interested in this topic ourselves. Alas, many papers did not report these additional information, which exacerbated the aforementioned problem even further. To better highlight this problem we revised the conclusions portion of the manuscript accordingly.

To me, one of the more interesting results of the analysis is confirmation of a scale effect that reveals itself, precisely, when one throws data from varied sites and settings into a single basket, as do the authors. It would therefore behove the authors to take the additional step of attempting to fit a truncated power variogram to their data, thereby potentially showing that such a variogram captures a greater range of behaviors with fewer parameters than do the stationary variograms they consider. Not only might a truncated power variogram prove to be more general and parsimonious than do standard stationary variogram, but they would be much more suitable than the latter when one needs to predict, using a transport model, how a plume of contaminant might migrate and behave outside the domain originally used to define the variogram.

I think that with additional work along the lines I have just proposed, this paper could stand out as a much more significant contribution to the literature than it does now.

We agree that using the available data set with a truncated power variogram can help to investigate the scaling behavior of subsurface media indicated by a number of different studies. We, therefore, re-did the analysis using this variogram function. Our results show that a truncated power law variogram achieves a similar goodness of fit compared to the other variogram functions. Depending on the application, such a lack of scale dependency can be considered an asset. We also investigated the behavior of the Hurst coefficient and critically discussed our findings. We revised the manuscript and included these results and some discussion on this topic into the manuscript.

---

## Author Response (AR2)

**reply:hess-2023-15-MS**

**July 2023**

Dear authors, The paper is fine and I like it (see the bottom part of this document for some very minor suggestions). But I had severe difficulties when makin sure that the data repository is accessible. T

Dear Jesus,

thank you again for editing our manuscript. Please find attached the revised version of the manuscript and our responses. We hope that the revisions to our manuscript and our responses address your wishes and look forward to hearing from you.

Yours, Falk

- he links in the paper do not work properly (check them in an "untrained" computer, i.e., any non-UFZ computer). In fact, the link in line 581 leads to https://www.copernicus.org/ Anyway, since the links led me to invalid webpages, I tried to search directly in Github.com. When I seach for "geostatDB", I get "This is the first release of the geostatDB package. geostatDB is an R package that provides access to the World Wide Hydrological Parameters DAtabase (WWHYPDA). When I seach for "geostatDB", "GeoStat-Examples/GeoStat-DB", or "GeoStat-Examples" I get no repository found Finally, I "accidentally" got to the GeoStat-Examples repository when looking for GStools, but could not find the csv files. This discussion proves that I am a clumsy github user (largely a passive one). But I feel you should facilitate the work by providing the right indications on how to access the repository. Minor comments (take these as suggestions, you do not need to follow them).

The fact that you could not access the GitHub repository is indeed a problem since the data collected for this paper is one of its main assets. However, when we tried the link, it worked even on non-UFZ computers. To be absolutely sure that data can be accessed we made the link to the Zenodo repository better available. As we already comment in the "Code and data availability" section, the data on the GitHub repository are subject to change as more data become available. The Zenodo link on the other hand provides a compressed archived of the Python scripts and csv data files as they were used for this manuscript. They do provide a snapshot of the GitHub repository as it was at the time the manuscript was submitted.

- Line 15: "soils" may perhaps be better than "soils sites"
  We changed this throughout the manuscript.

- Line 69: You may drop "In the next section,"
  We changed this as suggested.

- Line 75: "we critically" better than "we will critically"
  We changed this as suggested.

- Line 144: With your definitions (your definition of sigma2 is awkward to me), it is not correct C(h)=sigma2 rho(h), as C(0)=s. You may either point at the discontinuity of C(h) at the origin or, perhaps simpler, write C(h)=s-gamma(h)
  We decided for $C(h) = s - \gamma(h)$ and changed it accordingly in the manuscript.

- I am somewhat disappointed that you did not follow the advice of referee 1 (which I supported in the decision letter) of acknowledging that since the variogram is usually the only reference to geology in geostat analyses. Your paper confirms beautifully that data does not suffice to choose the variogram. It is geology what does, which is why structural analysis is the most important step in variogram estimation. This is essential for the behavior at the origin (which you hardly discuss) and for the range (which you do). I accept that, for discussion purposes, you take the maximum length scale as prior. But, for each site, you should use the variability scale that geology tells you. In fact, this is the one I have always used rather than fitting to the sample variogram (for which there is no convergence proof). This is particularly true for the vertical direction (as proven by your discussion in lines 355-360). And this deserves a discussion.
  We agree that this point and its ramifications deserves a dedicated discussion in the manuscript. As a result, we added a paragraph to the section discussing the limitations and possible pitfalls of our analysis. In this paragraph, we stress how a purely data-drive analysis cannot be a substitute for good domain knowledge and site-specific expertise in the determination of an appropriate variogram model and successful geostatistical investigation.

- Please, check figure 8. It looks like three, rather than two, Gaussians.
  We agree that the kernel-density estimate may be even better approximated using a mixture model of three Gaussians. However, our aim here was to show that even with a comparably simple model, an excellent fit could be achieved. We revised the accompanying text to better convey this point.

- Line 365: The distinction between "measurement errors and unresolved variations in the measured variable below the measurement scale" is fundamental both conceptually (again, importance of structural analysis) and practically (Kriging and simulations are different). While not affecting

your analysis, you should revise the statement.
We revised the statement to make clear that the two causes are very different.